# SEE-OoD: Supervised Exploration For Enhanced Out-of-Distribution Detection

## Abstract

Current techniques for Out-of-Distribution (OoD) detection predominantly rely on quantifying predictive uncertainty and incorporating model regularization during the training phase, using either real or synthetic OoD samples. However, methods that utilize real OoD samples lack exploration and are prone to overfit the OoD samples at hand. Whereas synthetic samples are often generated based on features extracted from training data, rendering them less effective when the training and OoD data are highly overlapped in the feature space. In this work, we propose a Wasserstein-score-based generative adversarial training scheme to enhance OoD detection accuracy, which, for the first time, performs *data augmentation* and *exploration* simultaneously under the *supervision* of limited OoD samples. Specifically, the generator explores OoD spaces and generates synthetic OoD samples using feedback from the discriminator, while the discriminator exploits both the observed and synthesized samples for OoD detection using a predefined Wasserstein score. We provide theoretical guarantees that the optimal solutions of our generative scheme are statistically achievable through adversarial training in empirical settings. We then demonstrate that the proposed method outperforms state-of-the-art techniques on various computer vision datasets and exhibits superior generalizability to unseen OoD data.

## 1 Introduction

Deep Neural Networks (DNNs) have been recently deployed in various real applications demonstrating their efficacious capacities in learning inference tasks, such as classification (He et al., 2016; Huang et al., 2016), object detection (Girshick, 2015; Redmon et al., 2016), and machine translation (Tan et al., 2020; Zhang & Zong, 2020). Most of these tasks, however, assume that training and testing samples have the same data distribution (Krizhevsky et al., 2017; He et al., 2015; Drummond & Shearer, 2006) under which DNN models are trained in a closed-world manner (Yang et al., 2021). This assumption might not hold in practical applications where control over testing samples is limited. Several researchers have relaxed the former statement by assuming that testing samples can be essentially different from samples in the training distribution. We refer to those testing samples as OoD samples (i.e. **O**ut-**o**f-**D**istribution) whereas those coming from the training data distribution as InD samples (i.e. **In-D**istribution). This motivates the problem of training DNNs that can effectively classify InD samples while simultaneously detecting OoD samples. One practical application arises in self-driving vehicles (Tambon et al., 2022; Yang et al., 2021) for which a reliable DNN control system is expected to identify scenarios that are far from what has been observed during training stages and prompt warning to the driver rather than blindly react to them. This renders OoD detection crucial for reliable machine learning models in real-world applications. In this paper, we focus on solving the problem of training DNN classifiers that can effectively identify OoD samples while maintaining decent classification performance for InD data.

Most existing works on OoD detection for DNN models leverage the predictive uncertainty of the pre-trained DNNs to separate InD and OoD samples in a predefined score space (Liang et al., 2020; Lee et al., 2018; Hendrycks & Gimpel, 2016; Liu et al., 2020). In particular, these methods adopt score functions that quantify the uncertainty of the predictions and project these scores to different extrema in the score continuum, representing low and high predictive uncertainty, respectively. For instance, Hendrycks & Gimpel (2016) retrieved the maximum softmax probability (MSP) among all classes as the uncertainty score for an incoming sample whereas Liu et al. (2020) utilized the energy

Table 1: Summary of OoD detection methods

| Method Family | Examples | Score Function | OoD-aware Training | Real OoD | OoD Space Exploration | Theoretical Justification |
|---|---|:---:|:---:|:---:|:---:|:---:|
| *Calibration* | MSP (Hendrycks & Gimpel, 2016); ODIN (Liang et al., 2020); Maha (Lee et al., 2018); Energy (Liu et al., 2020) | ✓ | ✗ | ✗ | ✗ | ✗ |
| *Virtual outlier generation* | VOS (Du et al., 2022); GAN-Synthesis (Lee et al., 2017) | ✓ | ✓ | ✗ | ✓ | ✗ |
| *OoD-based methods* | Energy + Finetune (Liu et al., 2020); WOOD (Wang et al., 2021) | ✓ ✓ | ✓ ✓ | ✓ ✓ | ✗ ✗ | ✗ ✓ |
| ***Guided OoD exploration*** | **SEE-OoD** | ✓ | ✓ | ✓ | ✓ | ✓ |

score of samples to achieve InD/OoD separations. To extract more information from the pre-trained models and reduce unnecessary noises, Liang et al. (2020) calibrated the output probability by temperature scaling (Hinton et al., 2015; Pereyra et al., 2017). Lee et al. (2018), however, operated directly on the features and defined the confidence score based on Mahalanobis distances. With a well-calibrated score function, such methods can perform OoD detection on pre-trained DNNs by simply adding an additional module without the need for re-training.

Despite being computationally efficient, these *calibration-based methods* only operate in the inference phase by manipulating the output of the pre-trained models, whose parameters are already fixed after training. This may result in relatively poor performance as they fail to exploit the capacity of DNNs in InD/OoD separation tasks. One potential approach for resolving this issue is to incorporate OoD detection in the training objective and regularize the classifier in the training stage using *virtual outliers* generated based on InD data. For instance, Lee et al. (2017) used GANs (Goodfellow et al., 2020) to generate InD boundary samples and proposed a training scheme that jointly optimizes the classification objective and retains a model less confident about the generated virtual outliers. Similarly, Du et al. (2022) modeled InD data as a multivariate Gaussian distribution and sampled virtual outliers from their tails. These samples are then used in a regularization framework for classification and OoD detection. A major drawback of such methods is that the generation of boundary outliers is heavily coupled with the features learned on InD data. This arises when InD and OoD data are heavily overlapped in feature spaces. In such scenarios, generating outliers purely based on low-density features without any supervision from real OoD data can return virtual outliers that are not good representatives of real OoD samples.

Avoiding the issue of unsupervised generation of OoD samples, several works have studied problem instances in which empirical knowledge about OoD data is available. In fact, many real applications allow for identifying potential OoD samples based on training data. For instance, in face recognition applications (Yu et al., 2020), it is reasonable to assume that images with no human faces are OoD data. In such settings, several methods that exploit given OoD samples to learn InD/OoD separation while training for classification were proposed. We refer to such methods that are directly trained or fine-tuned on both InD and real OoD data as *OoD-based methods*. For example, Liu et al. (2020) fine-tuned a pre-trained model on real OoD data to achieve InD/OoD separation in the energy score space. More recently, Wang et al. (2021) proposed the WOOD detector, which uses a Wasserstein-distance-based (Rüschendorf, 1985; Villani, 2008) score function and is directly trained on both InD and OoD data to map them to high and low confidence scores, respectively. With a sufficient training OoD sample size, WOOD (Wang et al., 2021) achieves state-of-the-art OoD detection performance on multiple benchmark experiments on computer vision datasets.

A major limitation for existing *OoD-based methods* is that learning such InD/OoD score mapping can be challenging when the number of real OoD samples in training is limited. In such cases, the model is prone to over-fit OoD data samples which can result in low OoD detection accuracy for unseen data. One plausible solution is to combine *OoD-based methods* with data augmentation techniques like transformation and perturbation (Shorten & Khoshgoftaar, 2019; Lemley et al., 2017). Although data augmentation can mitigate the over-fitting problem of these *OoD-based methods*, the augmented data can still suffer from a poor representation of the OoD space. Table 1 provides a thorough overview of the drawbacks and advantages of each of the aforementioned method.

Motivated by these drawbacks, we propose a generative adversarial approach that utilizes real OoD data for supervised generation of OoD samples that can better explore the OoD space. Our proposed approach tackles the two drawbacks of existing methods, that is, improves *virtual outlier generation methods* by utilizing real OoD samples for a supervised OoD generation scheme; and simultaneously augments OoD data with exploration to overcome the issue of poor and insufficient OoD samples in *OoD-based methods*. The main idea is to *iteratively exploit OoD samples to explore OoD spaces using feedback from the model*. Specifically, we introduce a **S**upervised-**E**xploration-based generative adversarial training approach for **E**nhanced **O**ut-**o**f-**D**istribution (**SEE-OoD**) detection, which is built on the Wasserstein-score function (Wang et al., 2021). The generator is designed to explore potential OoD spaces and generate virtual OoD samples based on the feedback provided by the discriminator, while the discriminator is trained to correctly classify InD data and separate InD and OoD in the Wasserstein score space. Our contributions can be summarized as the following:

- We propose a Wasserstein-score-based (Wang et al., 2021) generative adversarial training scheme where the generator explores OoD spaces and generates virtual outliers with the feedback provided by the discriminator, while the discriminator exploits these generated outliers to separate InD and OoD data in the predefined Wasserstein score space. (Sec. 2.2)

- We provide several theoretical results that guarantee the effectiveness of our proposed method. We show that at optimality, the discriminator is expected to perfectly separate InD and OoD (including generated virtual OoD samples) in the Wasserstein score space. Furthermore, we establish a generalization property for the proposed method. (Sec. 2.3)

- We introduce a new experimental setting for evaluating OoD detection methods: *Within-Dataset* OoD detection, where InD and OoD are different classes of the same dataset, and is a more challenging task for DNNs compared to the commonly used *Between-Dataset* OoD separation tasks (Liang et al., 2020; Wang et al., 2021). We then demonstrate the effectiveness of our method on multiple benchmark experiments with different settings on image datasets. (Sec. 3)

## 2 METHODOLOGY

We present our method for OoD detection under the supervised classification framework, where a well-trained neural network model is expected to correctly classify InD data while effectively identifying incoming OoD testing samples. In general, we denote the distributions for InD data and OoD data as $\mathbb{P}_{\text{InD}}(\mathbf{x}, y)$ and $\mathbb{P}_{\text{OoD}}(\mathbf{x})$, where $\mathbf{x}$ and $y$ represent inputs and labels, respectively. Note that for OoD data, we only have the marginal distribution of inputs $\mathbb{P}_{\text{OoD}}(\mathbf{x})$ as there are no labels for them. For simplicity purposes, we use $d$ to denote the dimension of inputs. For instance, in the context of computer vision, $\mathbf{x} \in \mathbb{R}^d$ is a flattened tensor of an image that has $C$ channels, $H$ pixels in height, and $W$ pixels in width. The corresponding dimension of inputs is $d = C \times H \times W$. Throughout this paper, the number of classes of InD data is denoted by $K$ and labels are represented by the set $\mathcal{K}_{\text{InD}} = \{1, ..., K\}$. Under this framework, the classic OoD detection problem is equivalent to finding a decision function $\mathcal{F} \colon \mathbb{R}^d \longmapsto \{0, 1\}$ such that:

$$\mathcal{F}(\mathbf{x}) = \begin{cases} 0, & (\mathbf{x}, y) \sim \mathbb{P}_{\text{InD}}(\mathbf{x}, y) \\ 1, & \mathbf{x} \sim \mathbb{P}_{\text{OoD}}(\mathbf{x}) \end{cases}, \tag{1}$$

where the decision function $\mathcal{F}(\cdot)$ can be constructed by combining a DNN classification model with well-defined score functions that return different values for InD and OoD data. In this paper, we follow previous literature (Yang et al., 2021; Liang et al., 2020; Du et al., 2022) and define OoD data as data that does not come from the training distribution (i.e. InD).

### 2.1 WASSERSTEIN-DISTANCE-BASED SCORE FUNCTION

In this work, we adopt the Wasserstein score function introduced by Wang et al. (2021) to quantify the uncertainty of the model predictions. Given a cost matrix $M \in \mathbb{R}^{K \times K}$ and a classification function $f \colon \mathbb{R}^d \longmapsto \mathbb{R}^K$ that maps an input sample to a discrete probability distribution of predictions, the Wasserstein score for an input sample $\mathbf{x}$ is defined by:

$$\mathcal{S}(f(\mathbf{x}); M) := \min_{k \in \mathcal{K}_{\text{InD}}} W(f(\mathbf{x}), \mathbf{e}_k; M) = \min_{k \in \mathcal{K}_{\text{InD}}} \inf_{P \in \Pi(f(\mathbf{x}), \mathbf{e}_k)} \langle P, M \rangle, \tag{2}$$

where $W(p_1, p_2; M)$ is the Wasserstein distance (Rüschendorf, 1985; Cuturi, 2013) between two discrete marginal probability distributions $p_1$ and $p_2$ under the cost matrix $M$, $\mathbf{e}_k \in \mathbb{R}^K$ is the $K$-dimensional one-hot vector where only the $k$th element is one, and $P$ is a joint distribution that belongs to the set of all possible transport plans $\Pi(f(\mathbf{x}), \mathbf{e}_k) := \{P \in \mathbb{R}_+^{K \times K} | P\mathbf{1}_K = \mathbf{e}_k, P\mathbf{1}_K^\top = f(\mathbf{x})\}$, where $\mathbf{1}_K$ is the all-one vector. In this work, we stick to the classic binary cost matrix $M_b$ (Frogner et al., 2015) where transporting an equal amount of probability mass between any two different classes yields the same costs; that is, $M_b = \mathbf{1}_{K \times K} - \mathbf{I}_K$ where $\mathbf{1}_{K \times K}$ is the all-ones matrix with dimension $K \times K$ and $\mathbf{I}_K$ is the $K$-dimensional identity matrix. Detailed descriptions on the definition of Wasserstein distance and cost matrix selection can be found in Appendix A.1.

**Remark 1.** *Under the binary cost matrix $M_b$, the Wasserstein score of an input sample $\mathbf{x} \in \mathbb{R}^d$ given a classifier function $f : \mathbb{R}^d \longmapsto \mathbb{R}^K$ is equivalent to $\mathcal{S}(f(\mathbf{x}); M_b) = 1 - \|f(\mathbf{x})\|_\infty$. Consequently, the minimum Wasserstein score is attained when $f(\mathbf{x})$ is any one-hot vector, reflecting its high predictive confidence, while the maximum is achieved when $f(\mathbf{x})$ outputs the same probability for each class, implying high predictive uncertainty.*

This justifies the reasons for using the Wasserstein score function to quantify the predictive uncertainty. For an ideal classifier, we expect InD samples to have lower Wasserstein scores, which indicates classifiers' high confidence when assigning them to one of the $K$ classes. In contrast, OoD samples should have higher scores, reflecting the high uncertainty of classifying them into any one of the classes. Then, given a cost matrix $M$, a well-trained classifier $f(\cdot)$, and a threshold $\eta$, the score-based detector for an incoming sample $\mathbf{x}$ can be formalized below in the same manner as in previous works (Liang et al., 2020; Wang et al., 2021):

$$\mathcal{F}(\mathbf{x}; f, M, \eta) = \mathbb{1}_{[\mathcal{S}(f(\mathbf{x}); M) > \eta]} = \begin{cases} 0, & \mathcal{S}(f(\mathbf{x}); M) \leq \eta \\ 1, & \mathcal{S}(f(\mathbf{x}); M) > \eta \end{cases} = \begin{cases} 0, & \|f(\mathbf{x})\|_\infty \geq 1 - \eta \\ 1, & \|f(\mathbf{x})\|_\infty < 1 - \eta \end{cases}, \quad (3)$$

where the last equality holds under the pre-defined $M_b$. The decision threshold $\eta \in [0, 1]$ is chosen to satisfy a pre-specified True Negative Rate (TNR) at the inference phase, which is defined as the proportion of InD samples that are correctly classified as InD by the detector. We next inherit the score function defined in Eq. (3) in an adversarially generative formulation for jointly training InD classification and InD/OoD separation.

## 2.2 SUPERVISED-EXPLORATION-BASED OUT-OF-DISTRIBUTION DETECTION

In this section, we introduce a Wasserstein-score-based generative adversarial scheme for training classification models that can detect OoD samples, where the generator aims at exploring the potential OoD spaces with the feedback provided by the discriminator, while the discriminator exploits the advantages of these generated points to separate InD and OoD samples. In this paper, we denote the discriminator as $D(\mathbf{x}; \theta_D)$ where it outputs a $K$-dimensional predicted discrete probability distribution for the input image $\mathbf{x}$. The generator is represented by $G(\mathbf{z}; \theta_G)$ where it maps an $n$-dimensional noise vector $\mathbf{z} \in \mathbb{R}^n$ that is drawn from some prior distribution $\mathbb{P}_\mathbf{z}$ to the data space. Note that $D$ and $G$ are essentially two different neural networks that are parameterized by $\theta_D$ and $\theta_G$, respectively. By convention, we assume that $\theta_D$ and $\theta_G$ belong to a subset of the unit ball (Arora et al., 2017). The overall minimax objective function for our method is as follows, where we slightly abuse the notation and use $D$ and $G$ without writing out their parameters explicitly,

$$\min_D \max_G \mathcal{L}(D, G) = \min_D \max_G \underbrace{\mathbb{E}_{(\mathbf{x}, y) \sim \mathbb{P}_{\text{InD}}(\mathbf{x}, y)} \left[ -\log(D(\mathbf{x})^\top \mathbf{e}_y) \right]}_{\text{(1) InD Classification}}$$

$$- \beta_{\text{OoD}} \underbrace{\mathbb{E}_{\mathbf{x} \sim \mathbb{P}_{\text{OoD}}(\mathbf{x})} \left[ \mathcal{S}(D(\mathbf{x}); M_b) \right]}_{\text{(2) OoD Wasserstein Score Mapping}} + \beta_z \underbrace{\mathbb{E}_{\mathbf{z} \sim \mathbb{P}_\mathbf{z}} \left[ \mathcal{S}(D(G(\mathbf{z})); M_b) \right]}_{\text{(3) OoD Adversarial Training}}, \quad (4)$$

where $\beta_{\text{OoD}}, \beta_z > 0$ are the hyperparameters that balance the losses of the generator and discriminator. In this paper, a multivariate Gaussian distribution with zero mean and identity covariance matrix $\mathbf{I}_n$ is chosen as the default prior. This minimax objective function can be understood and decomposed into two parts: (1) aims at training the discriminator to achieve high classification accuracy on InD data while simultaneously assigning low Wasserstein scores to them, while (2) and (3) together emulate the original GAN formulation (Goodfellow et al., 2020; Arjovsky et al., 2017) but in the Wasserstein score space, where $G$ and $D$ are trained to explore and generate virtual OoD samples while mapping OoD data to high Wasserstein scores. Unlike existing methods that generate outliers

without recourse to observed OoD data (Du et al., 2022; Lee et al., 2017), our method allows for the explorative generation of synthetic samples. In the iterative optimization process, the discriminator gradually learns the Wasserstein score mapping of InD and OoD samples, while the generator utilizes this knowledge as guidance to generate samples that retain a high Wasserstein score. Moreover, as the proposed SEE-OoD operates on the Wasserstein score space rather than the data space, the generated OoD samples do not necessarily resemble the target distribution (i.e. observed OoD) in the data space, which encourages our model to explore OoD spaces beyond the observed samples.

To solve the presented optimization problem, we propose an iterative algorithm that alternatively updates $D$ and $G$ using minibatch stochastic gradient descent/ascent outlined in Algorithm 1. After training, the discriminator $D$ is then utilized to construct a threshold-based decision function $\mathcal{F}(\mathbf{x}; D, M_b, \eta) = \mathbb{1}_{[\mathcal{S}(D(\mathbf{x}); M_b) > \eta]}$ for OoD detection. The decision threshold $\eta$ is chosen such that $\mathbb{E}_{\mathbf{x} \sim \mathbb{P}_{\mathrm{InD}}(\mathbf{x})} \mathcal{F}(\mathbf{x}; D, M_b, \eta) = \alpha$, with $1 - \alpha \in [0, 1]$ representing the probability that an incoming InD sample is correctly identified as InD by the detector (i.e. TNR).

## 2.3 THEORETICAL RESULTS

In this section, we provide theoretical guarantees that demonstrate the effectiveness of our method.

**Theorem 1.** *For a given discriminator $\bar{D}$, let $G_{\bar{D}}^\star$ be the optimal solution among all possible real-valued functions that map $\mathbb{R}^n$ to $\mathbb{R}^d$, then the Wasserstein scores of the generated data are lower bounded by the Wasserstein scores of OoD data, that is,*

$$\mathbb{E}_{\mathbf{z} \sim \mathbb{P}_{\mathbf{z}}} \left[ \mathcal{S}(\bar{D}(G_{\bar{D}}^\star(\mathbf{z})); M_b) \right] \geq \mathbb{E}_{\mathbf{x} \sim \mathbb{P}_{OoD}(\mathbf{x})} \left[ \mathcal{S}(\bar{D}(\mathbf{x}); M_b) \right]. \tag{5}$$

Theorem 1 guarantees that for any discriminator $D$, the generated synthetic data at optimal $G$ retain desired high Wasserstein scores. We next show that at optimality, the discriminator perfectly classifies the InD data and separates InD and OoD data in the Wasserstein score space.

**Theorem 2.** *Let $D$ and $G$ belong to the sets of all possible real-valued functions, in particular, neural networks, such that $D : \mathbb{R}^d \longmapsto \mathbb{R}^K$ and $G : \mathbb{R}^n \longmapsto \mathbb{R}^d$, respectively. Then, under optimality, $D^\star$ and $G^\star$ possess the following properties:*

$$D^\star(\mathbf{x}) = \begin{cases} \mathbf{e}_y, & (\mathbf{x}, y) \sim \mathbb{P}_{InD}(\mathbf{x}, y) \\ \frac{1}{K} \mathbf{1}_K, & \mathbf{x} \sim \mathbb{P}_{OoD}(\mathbf{x}) \end{cases} \quad and \quad G^\star \in \{G : D^\star(G(\mathbf{z})) = \frac{1}{K} \mathbf{1}_K, \forall \mathbf{z} \sim \mathbb{P}_{\mathbf{z}}\}. \tag{6}$$

*Furthermore, if the discriminator $D$ is $\alpha$-Lipschitz continuous with respect to its inputs $\mathbf{x}$, where the Lipschitz constant $\alpha > 0$. Then, at optimality, $G^\star(\mathbf{z}) \not\sim \mathbb{P}_{InD}(\mathbf{x})$, $\forall \mathbf{z} \in \mathbb{P}_{\mathbf{z}}$; that is, the probability that the generated samples are In-Distribution is zero.*

**Remark 2.** *In practice, these optimal solutions can be obtained in over-parameterized settings. The purpose of these theoretical results is to give intuition on the dynamics of our min-max objective.*

Note that we use the notation $\mathbf{x} \not\sim \mathbb{P}_0$ to say that $f_0(\mathbf{x}) = 0$, where $f_0$ is the corresponding probability density function of $\mathbb{P}_0$. Theorems 1 and 2 assure that, at optimality, $G$ generates samples with high Wasserstein scores that do not belong to InD. These promising properties ensure that our generated OoD samples never overlap with InD samples in the data space, which does not hold in previous works on *virtual outlier generation* (Du et al., 2022; Lee et al., 2017). Therefore, the synthetic OoD samples generated by our model will only enhance the discriminator's understanding of the OoD space without undermining its classification performance in the InD space.

We now provide a generalization result that shows that the desired optimal solutions provided in the Theorems 2 can be achieved in empirical settings. Motivated by the *neural network distance* introduced by Arora et al. (2017) to measure the difference between the real and generated distributions in GANs, we define a generalized *neural network loss* for the proposed generative adversarial training framework, which quantifies the loss of the outer minimization problem for three distributions under a given set of measuring functions and can be easily generalized to a family of objective functions. Examples on the applications of *neural network loss* can be found in Appendix B.3.

**Definition 1.** *Let $\mathcal{F} : \mathbb{R}^d \longmapsto \mathbb{R}^K$ be a class of functions that projects the inputs to a $K$-dimensional probability vector, such that $f \in \mathcal{F}$ implies $\mathbf{1}_K(\cdot) - f \in \mathcal{F}$. Let $\Phi = \{\phi_1, \phi_2, \phi_3 : \mathbb{R}^K \longmapsto \mathbb{R}\}$ be a set of convex measuring functions that map a probability vector to a scalar score. Then, the neural network loss w.r.t. $\Phi$ among three distributions $p_1, p_2,$ and $p_3$ supported on $\mathbb{R}^d$ is defined as*

$$\mathcal{L}_{\mathcal{F}, \Phi}(p_1, p_2, p_3) = \inf_{D \in \mathcal{F}} \mathbb{E}_{\mathbf{x} \sim p_1} [\phi_1(D(\mathbf{x}))] + \mathbb{E}_{\mathbf{x} \sim p_2} [\phi_2(D(\mathbf{x}))] + \mathbb{E}_{\mathbf{x} \sim p_3} [\phi_3(\mathbf{1}_K - D(\mathbf{x}))].$$

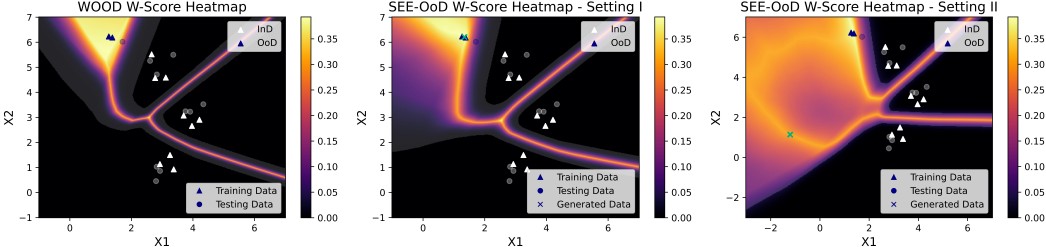

Figure 1: A 2D numerical illustration of the intuition behind SEE-OoD. In this figure, we present the Wasserstein score heatmaps of the WOOD (Wang et al., 2021) detector and *two* possible solution states of SEE-OoD after training, where brighter colors represent higher Wasserstein scores (i.e. OoD) and the shaded boundary is the InD/OoD decision boundary under 95% TNR. Details regarding this simulation study can be found in Appendix C.

For instance, in the context of SEE-OoD, three probability distributions $p_1, p_2$, and $p_3$ correspond to $\mathbb{P}_{\text{InD}}, \mathbb{P}_{\text{OoD}}$, and $\mathbb{P}_G$, respectively. With careful selection of measuring functions as introduced in Appendix B.3, the *neural network loss* recovers the outer minimization objective in Eq. (4) for a fixed $G$. The following Theorem 3 shows that the *neural network loss* generalizes well in empirical settings, and Corollary 1 guarantees that when taking the iterative training of $D$ and $G$ into account, the theoretical optima introduced in Theorem 2 is statistically achievable through training.

**Theorem 3.** *Let $p_1, p_2$, and $p_3$ be three distributions and $\widehat{p}_1, \widehat{p}_2$, and $\widehat{p}_3$ be the empirical versions with at least $m$ samples each. Suppose the measuring functions $\phi_i \in \Phi$ are $L_{\phi_i}$-Lipschitz continuous and take values in $[l_i, u_i]$ for $i \in \{1, 2, 3\}$. Let the discriminator $D$ be $L$-Lipschitz continuous with respect to its parameter $\theta_D$ whose dimension is denoted by $p$. Then, there exists a universal constant $C$ such that when the empirical sample size $m \geq \max_i \left\{ \frac{Cp(u_i - l_i)^2 \log(LL_{\phi_i} p/\epsilon)}{\epsilon^2} \right\}$, we have with probability at least $1 - exp(-p)$ over the randomness of $\widehat{p}_1, \widehat{p}_2$, and $\widehat{p}_3$,*

$$|\mathcal{L}_{\mathcal{F},\Phi}(p_1, p_2, p_3) - \mathcal{L}_{\mathcal{F},\Phi}(\widehat{p}_1, \widehat{p}_2, \widehat{p}_3)| \leq \epsilon \qquad (7)$$

**Corollary 1.** *In the setting of Theorem 3., suppose that $\{G^{(i)}\}_{i=0}^N$ be the $N$ generators in the $N-$ iterations of the training, and assume $\log N \leq p$ and $\log N \ll d$. Then, there exists a universal constant $C$ such that when $m \geq \max_i \left\{ \frac{Cp(u_i - l_i)^2 \log(LL_{\phi_i} p/\epsilon)}{\epsilon^2} \right\}$, with probability at least $1 - exp(-p)$, for all $t \in [N]$,*

$$|\mathcal{L}_{\mathcal{F},\Phi}(\mathbb{P}_{InD}, \mathbb{P}_{OoD}, \mathbb{P}_{G^{(t)}}) - \mathcal{L}_{\mathcal{F},\Phi}(\widehat{\mathbb{P}}_{InD}, \widehat{\mathbb{P}}_{OoD}, \widehat{\mathbb{P}}_{G^{(t)}})| \leq \epsilon. \qquad (8)$$

## 2.4 NUMERICAL ILLUSTRATION

We provide a small-scale simulation study to visually illustrate our proposed approach. To shed light on the mechanisms underpinning our method, we specifically explore two distinct hyperparameter configurations, as depicted in Figure 1. In the first setting, the hyperparameter is chosen such that $\beta_{\text{OoD}} > \beta_z$ and $n_d > n_g$, leading to a *dominant* discriminator throughout the training process. We observe that after training, the discriminator assigns high Wasserstein scores only if the input $\mathbf{x} \in \mathbb{R}^2$ is close to the training OoD samples. In this case, the generator augments the limited OoD data by exploring regions close to them, therefore the proposed method can be understood as a WOOD detector (Wang et al., 2021) with our proposed explorative data augmentation. The middle panel in Figure 1 shows the Wasserstein score heatmap obtained under this setting, where the proposed SEE-OoD detector results in larger OoD rejection regions around OoD samples compared to the WOOD method, whose Wasserstein score heatmap is given by the left panel in Figure 1.

In the second setting, we set $\beta_{\text{OoD}} < \beta_z$ and $n_d < n_g$. In this scenario, the generator is *dominant* so it can fool the discriminator even when the generated data are not in the vicinity of the observed OoD samples. Thus, in the iterative training process, the generator keeps exploring OoD spaces, while the discriminator learns to project more regions, that the generator has explored, to high Wasserstein scores. This case is demonstrated by the right panel of Figure 1, where the generated samples are far away from observed OoD samples and the OoD region is larger than that of WOOD. This demonstrates the effectiveness of exploration and the advantages of the proposed generative

---

**Algorithm 1: SEE-OoD:** $B_{\text{InD}}, B_{\text{OoD}},$ and $B_{\text{G}}$ are minibatch sizes; $\beta_{\text{OoD}}, \beta_z \in \mathbb{R}_+$ are regularization parameters; $n \in \mathbb{Z}_+$ is the dimension of the Gaussian noise; $\eta_D, \eta_G \in \mathbb{R}_+$ are learning rates, $n_d, n_g \in \mathbb{Z}_+$ and the number of ascent/descent updates for $D$ and $G$ per iteration.

---

**for** number of training iterations **do**

    **for** number of discriminator updates $n_d$ **do**

        • Randomly sample $B_{\text{InD}}$ InD samples $\{(\mathbf{x}_{\text{InD}}^{(i)}, y_{\text{InD}}^{(i)})\}_{i=1}^{B_{\text{InD}}}$ from $\mathbb{P}_{\text{InD}}(\mathbf{x}, y)$.

        • Randomly sample $B_{\text{OoD}}$ OoD samples $\{\mathbf{x}_{\text{OoD}}^{(i)}\}_{i=1}^{B_{\text{OoD}}}$ from $\mathbb{P}_{\text{OoD}}(\mathbf{x})$.

        • Generate $B_{\text{G}}$ samples $\{G(\mathbf{z}^{(i)})\}_{i=1}^{B_{\text{G}}}$, where $\mathbf{z}^{(i)} \sim \mathcal{N}(\mathbf{0}_n, \mathbf{I}_n), \forall i \in \{1, ..., B_{\text{G}}\}$.

        • Forward propagation to compute $\mathcal{L}(D, G)$ and update $D$ by stochastic gradient descent:

$$\theta_D \longleftarrow \theta_D - \eta_D \nabla_{\theta_D} \left[ \frac{1}{B_{\text{InD}}} \sum_{i=1}^{B_{\text{InD}}} -\log D(\mathbf{x}_{\text{InD}}^{(i)\top}) \, \mathbf{e}_{y_{\text{InD}}^{(i)}} \right.$$

$$\left. - \beta_{\text{OoD}} \frac{1}{B_{\text{OoD}}} \sum_{i=1}^{B_{\text{OoD}}} \mathcal{S}(D(\mathbf{x}_{\text{OoD}}^{(i)}); M_b) + \beta_z \frac{1}{B_{\text{G}}} \sum_{i=1}^{B_{\text{G}}} \mathcal{S}(D(G(\mathbf{z}^{(i)})); M_b) \right]$$

    **end for**

    **for** number of generator updates $n_g$ **do**

        • Generate $B_{\text{G}}$ samples $\{G(\mathbf{z}^{(i)})\}_{i=1}^{B_{\text{G}}}$, where $\mathbf{z}^{(i)} \sim \mathcal{N}(\mathbf{0}_n, \mathbf{I}_n), \forall i \in \{1, ..., B_{\text{G}}\}$.

        • Forward propagation on current $D$ and update $G$ by stochastic gradient ascent:

$$\theta_G \longleftarrow \theta_G + \eta_G \nabla_{\theta_G} \left[ \beta_z \frac{1}{B_{\text{G}}} \sum_{i=1}^{B_{\text{G}}} \mathcal{S}(D(G(\mathbf{z}^{(i)})); M_b) \right]$$

    **end for**

**end for**

---

adversarial scheme over naive *OoD-based methods*. Here, the generated data shown in the figure only reflects the final state of the generator after training. In fact, the generator will generate different OoD samples according to the feedback provided by the discriminator in each iteration.

That said, it should be noted that the dynamics between the discriminator and generator are difficult to control through hyperparameter manipulation when dealing with real-world datasets. Indeed, the choice of the parameters is often dataset-dependent. Nevertheless, this numerical simulation aims to provide insights into the mechanisms behind our method. We will showcase in the next section that the proposed method achieves state-of-the-art OoD detection and generalization performance on a wide variety of real dataset experiments.

## 3 EXPERIMENTAL RESULTS

To demonstrate the effectiveness of the proposed SEE-OoD, we conducted several experiments and compared the results to state-of-the-art baseline methods. Our experiments considered various computer vision datasets, including MNIST (LeCun & Cortes, 2010), FashionMNIST (Xiao et al., 2017), CIFAR-10 (Krizhevsky, 2009), and SVHN (Netzer et al., 2011). We divide the experiments into two types: (1) *Between-Dataset* separation, where InD and OoD data are sampled from two different datasets; and (2) *Within-Dataset* separation, where InD and OoD data are sampled from different classes in the same dataset. The setting in the second task is closer to real-world scenarios and makes the OoD detection task more challenging as data from the same dataset are generally expected to be more akin to each other. For example, for defect classification systems in manufacturing, a potential OoD sample can be an unknown type of defect that did not show up in training but possesses similar features as those pre-known defects. Details of InD and OoD dataset pairs used for various experiments can be found in Table 2. We also test our methods in two possible real scenarios: (I) the observed OoD data is *balanced* (i.e. all OoD classes are observed and each class has a comparable amount of samples) and (II) the observed OoD data is *imbalanced* (i.e. only few classes are observed). Specifically, the first regime corresponds to cases with good empirical knowledge of OoD

Table 2: InD/OoD dataset pair configuration. Note that for *Within-Dataset* type experiment, a dataset is split into InD and OoD based on the labels specified in the table.

| Type | Experiment Dataset | InD Dataset | OoD Dataset | Training Sample Size (InD) | Testing Sample Size (InD/OoD) |
|---|---|---|---|---|---|
| *Within-Dataset* | MNIST | [2,3,6,8,9] | [1, 7] | 29807 | 4983/2163 |
| | FashionMNIST | [0, 1, 2, ..., 7] | [8, 9] | 48000 | 8000/2000 |
| | SVHN | [0, 1, 2, ..., 7] | [8, 9] | 63353 | 22777/3255 |
| *Between-Dataset* | MNIST-FashionMNIST | MNIST | FashionMNIST | 60000 | 10000/10000 |
| | CIFAR-10-SVHN | CIFAR-10 | SVHN | 60000 | 10000/26032 |

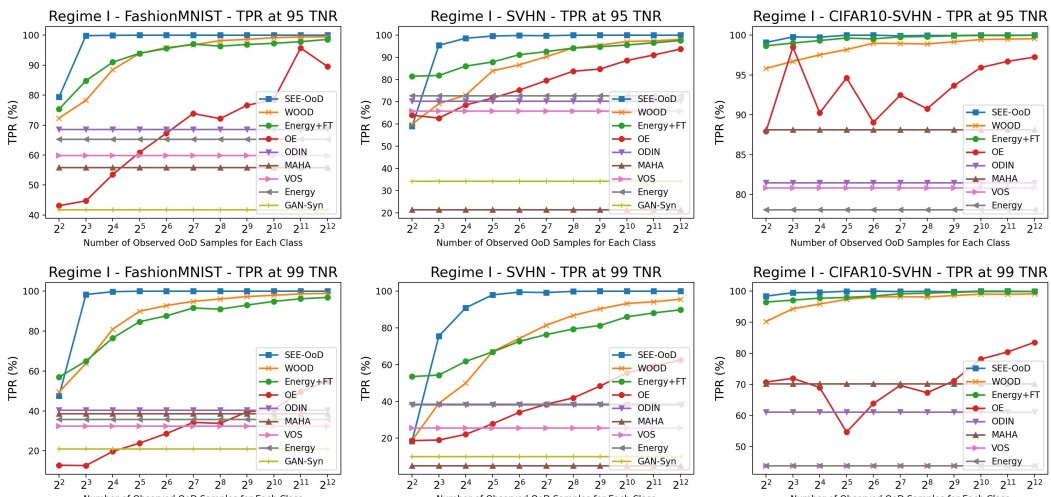

Figure 2: Results for Regime I experiments. The first row shows the TPR (i.e. detection accuracy) under 95% TNR for three of five experiments introduced in Table 2, whereas the second row shows the TPR under 99% TNR. Complete results can be found in Appendix E.

data but limited samples, whereas the second regime imitates the setting where neither the empirical knowledge nor the samples are sufficient.

We select the state-of-the-art classification network DenseNet (Huang et al., 2016) as the backbone model for the discriminator and design the generator with a series of transposed convolution blocks (Long et al., 2015; Radford et al., 2015; Noh et al., 2015). Details about model architectures and training hyperparameters can be found in Appendix D. In all experiments, the number of training OoD samples is carefully controlled and increased gradually in order to understand the difference between *OoD-based methods* and the proposed SEE-OoD. We then report the True Positive Rate (TPR) of OoD detection at 95% (or 99%) True Negative Rate (TNR), which is interpreted as the probability that a positive sample (i.e. OoD) is classified as OoD when the TNR is as high as 95% (or 99%). In addition, we conduct three Monte Carlo replications to investigate the randomness and instability that are commonly found in adversarial training and calculate the mean of absolute deviation (MAD) of the metrics to quantify the methods' robustness.

### 3.1 REGIME I: OBSERVING *Balanced* OOD SAMPLES

Under the first regime, all OoD classes are observed in the training stage and the OoD training set is aggregated by sampling an equal amount of data from each OoD class. We notice that in both *Between-Dataset* and the more challenging *Within-Dataset* OoD detection tasks, the proposed SEE-OoD detector and those *OoD-based* methods (i.e. WOOD (Wang et al., 2021) & Energy Finetune (EFT) (Liu et al., 2020)) achieve better performance than the methods that rely on calibration and virtual outlier generation. This makes sense as introducing real OoD data provides more information to the training process and allows it to be done in a supervised manner. Figure 2 presents the experimental results for Regime I experiments, and it is clear that the proposed SEE-OoD outperforms WOOD and EFT significantly in all three settings. We also find that as more OoD samples

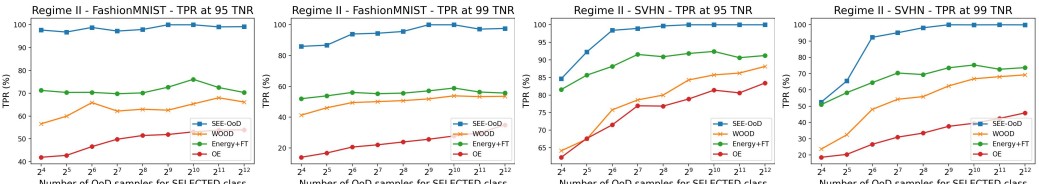

Figure 3: Results for Regime II experiments. For both experiments, only class 8 of the OoD classes is observed during the training stage, and the TPR under both $95\%$ and $99\%$ TNR are reported. Complete experimental results can be found in Appendix E.

are included in the training stage, the performance (i.e. TPR) of the proposed detector increases at a faster rate compared to other methods, implying that the proposed method utilizes and exploits the OoD samples in a more effective way. For instance, in the FashionMNIST *Within-Dataset* experiments, we identify that the proposed SEE-OoD achieves perfect separation (i.e. $100\%$ TPR) of InD/OoD when observing only $2^3$ samples for each class. In comparison, WOOD and EFT detectors can not achieve comparable detection accuracy, even with $2^{13}$ samples for each class, which also indicates that there is a performance cap for WOOD and EFT. One potential justification is that these methods only focus on observed OoD data without exploring the OoD spaces. Furthermore, as we start tolerating fewer false positives (i.e. higher TNR), the advantages of the proposed SEE-OoD are more obvious, implying that the decision boundary learned by SEE-OoD is tighter and the score distributions between InD and OoD are more separable.

## 3.2 REGIME II: OBSERVING *Imbalanced* OOD SAMPLES

In Regime II, we conduct *Within-Dataset* experiments on FashionMNIST and SVHN datasets, where only OoD samples from class 8 are provided in the training stage. Recall that in these two experiments, the OoD classes are both 8 and 9 (see Table 2 for details). In the inference stage, samples from both classes will be presented for testing, and an OoD detector with good generalization power is expected to not only identify samples from the seen class (i.e. class 8) but also those from the unseen OoD class (i.e. class 9) as OoD data. In Figure 3, we observe that in both experiments, the proposed SEE-OoD detector demonstrates a significant performance gain over the baseline methods. One can also observe that for baseline methods, observing more OoD samples in the training stage no longer benefits the detector after a certain point. For instance, in the SVHN experiments, the proposed SEE-OoD achieves nearly perfect TPR under the $95\%$ TNR whenever $2^6$ or more OoD samples are observed. In comparison, the detection performance of WOOD and EFT stops increasing with respect to the OoD sample size after reaching about $85\%$ and $91\%$ TPR, respectively. This bottleneck was not encountered in Regime I as both OoD classes 8 and 9 were observed. Our experiments show that while baseline methods suffer from lower detection performance when OoD classes are missing during training, our proposed method can still achieve near-perfect detection in the presence of sufficient OoD samples. This comparison confirms that the SEE-OoD detector benefits from the iterative exploration of OoD spaces in the training phase and exhibits better generalizability than baselines that are trained or finetuned solely based on existing OoD data.

## 4 CONCLUSIONS

In this paper, we propose a Wasserstein-score-based generative adversarial training scheme to enhance OoD detection. In the training stage, the proposed method performs data augmentation and exploration simultaneously under the supervision of existing OoD data, where the discriminator separates InD and OoD data in the Wasserstein score space while the generator explores the potential OoD spaces and augments the existing OoD dataset with generated outliers. We also develop several theoretical results that guarantee that the optimal solutions are statistically achievable in empirical settings. We provide a numerical simulation example as well as a comprehensive set of real-dataset experiments to demonstrate that the proposed SEE-OoD detector achieves state-of-the-art performance in OoD detection tasks and generalizes well towards unseen OoD data. The idea of *exploration with supervision* using generative models with feedback from OoD detectors creates many possibilities for future research in Out-of-Distribution learning.

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

# A WASSERSTEIN-DISTANCE-BASED SCORE FUNCTION

## A.1 DEFINITION OF WASSERSTEIN DISTANCE

**Definition A.1.** *(Wasserstein Distance restated) Let $p_1$ and $p_2$ be two $K$-dimensional discrete marginal probability distributions, then the Wasserstein Distance between them is defined as*

$$W(p_1, p_2; M) = inf_{P \in \Pi(p_1, p_2)} \langle P, M \rangle,$$

*where $\langle \cdot, \cdot \rangle$ denotes the Frobenius dot-product, $P$ is a joint distribution of $p_1$ and $p_2$ that belongs to the set of all possible transport plans $\Pi(p_1, p_2) := \{P \in \mathbb{R}_+^{K \times K} | P\mathbf{1}_K = p_2, P\mathbf{1}_K^\top = p_1\}$, and $M$ is the cost matrix that specifies the transport cost between any two classes and $\mathbf{1}_K$ is the $K$-dimensional all-one vector.*

## A.2 INTUITION BEHIND WASSERSTEIN-DISTANCE-BASED SCORE

The intuition behind using a score function is to quantify the predictive uncertainty of the output of DNN classification models, assuming that the difference between output probability distributions reflects the dissimilarities of the inputs. Hence the first step in tackling OoD problems is to find a good quantification of the dissimilarity between the output probability distributions. In this paper, the Wasserstein distance between discrete probability distributions, also known as the optimal transport distance (Villani, 2008; Cuturi, 2013), is utilized to measure the dissimilarity of two different distributions and construct the score function. Compared to Jensen-Shannon (JS) and Kullback-Leibler (KL) divergence, the Wasserstein distance is known to be more compatible with gradient-based methods because of its smoothness (Weng, 2019; Wang et al., 2021) and is widely used in previous literature in generative models and OoD detection (Villani, 2008; Wang et al., 2021; Arjovsky et al., 2017; Cuturi, 2013).

## A.3 COST MATRIX SELECTION

The cost matrix of the optimal transport problem under a discrete setting specifies the cost to transform probability mass from one class to another class (Villani, 2008; Wang et al., 2021), and hence its selection is usually application-specific and sometimes relies on good empirical knowledge. In this paper, we tackle the problem of OoD detection under a supervised classification framework and use computer vision datasets for which different classes within the dataset are treated with equal importance. Therefore, there is no reason to choose cost matrix other than the binary cost matrix $M_b$, where the transition of the same amount of probability density between any two different classes yields the same cost.

However, for real-world applications, users may hold valuable empirical knowledge about the relationship between different classes in the InD datasets. For example, for sign classification tasks that are common in self-driving cars, it is common to assume that a sign with the label *crosswalk ahead* is relatively closer to a sign with label *stop sign ahead* than to a sign with label *highway*, resulting in a lower transport cost. This motivates engineers to select their own cost matrix when utilizing the Wasserstein distance or deploying the proposed method into real-world industries according to their understanding of the applications.

# B OMITTED REMARKS AND PROOFS

## B.1 PROOFS FOR REMARK 1.

**Remark B.1.** *(Remark 1 restated) Under the binary cost matrix $M_b$, the Wasserstein score of an input sample $\mathbf{x} \in \mathbb{R}^d$ given a classifier function $f : \mathbb{R}^d \longmapsto \mathbb{R}^K$ is equivalent to $\mathcal{S}(f(\mathbf{x}); M_b) = 1 - \|f(\mathbf{x})\|_\infty$. Consequently, the minimum Wasserstein score is attained when $f(\mathbf{x})$ is any one-hot vector, reflecting its high predictive confidence, while the maximum is achieved when $f(\mathbf{x})$ outputs the same probability for each class, implying high predictive uncertainty.*

*Proof.* We first identify that the only possible joint probability distribution $P$ that satisfies the condition $P \in \Pi(f(\mathbf{x}), \mathbf{e}_k) := \{P \in \mathbb{R}_+^{K \times K} | P\mathbf{1}_K = f(\mathbf{x}), P\mathbf{1}_K^\top = \mathbf{e}_k\}$ is the following:

$$P_k^\star = \left[\mathbf{0}_K, ..., \underbrace{f(\mathbf{x})}_{k\text{th column}}, ..., \mathbf{0}_K\right] \in \mathbb{R}_+^{K \times K},$$

where the $k$th column is the predicted class probability vector $f(\mathbf{x})$ and all other columns are $\mathbf{0}_K$. Then, the proposed Wasserstein score function can be rewritten as the following:

$$\mathcal{S}(f(\mathbf{x}); M) := \min_{k \in \mathcal{K}_{\text{InD}}} W(f(\mathbf{x}), \mathbf{e}_k; M) = \min_{k \in \mathcal{K}_{\text{InD}}} \langle P_k^\star, M \rangle.$$

Under binary cost matrix $M_b = \mathbf{1}_{K \times K} - \mathbf{I}_K$; the Frobenius dot-product between $M_b$ and $P_k^\star$ can be further reduced to $1 - f_k(\mathbf{x})$, where $f_k(\mathbf{x})$ is the $k$th element of the predicted probability vector $f(\mathbf{x})$. Therefore, the Wasserstein score function is equivalent to

$$\mathcal{S}(f(\mathbf{x}); M_b) = \min_{k \in \mathcal{K}_{\text{InD}}} \langle P_k^\star, M_b \rangle = \min_{k \in \mathcal{K}_{\text{InD}}} 1 - f_k(\mathbf{x}) = 1 - \|f(\mathbf{x})\|_\infty.$$

And as we know that the infinity norm of a probability vector is bounded, that is, $\frac{1}{K} \leq \|f(\mathbf{x})\|_\infty \leq 1$, it follows that the maximum Wasserstein score of $1 - \frac{1}{K}$ is attained when $f(\mathbf{x}) = \frac{1}{K}\mathbf{1}_K$, whereas the minimum is achieved when $f(\mathbf{x}) = \mathbf{e}_k, \ \forall k \in \mathcal{K}_{\text{InD}}$. This completes the proof. $\qquad \square$

## B.2 PROOFS FOR REMARK B.2

**Remark B.2.** *Given the binary cost matrix $M_b$, the Wasserstein score function $\mathcal{S} : \mathbb{R}^K \longmapsto \mathbb{R}$, which maps a predicted probability vector to a scalar score, is Lipschitz continuous w.r.t. its inputs.*

*Proof.* By Remark 1, we know that $\mathcal{S}(f(\mathbf{x}); M_b) = 1 - \|f(\mathbf{x})\|_\infty$. Let $\mathbf{u}, \mathbf{v} \in \mathbb{R}^K$ be two arbitrary probability distributions, then we have the following:

$$|\mathcal{S}(\mathbf{u}; M_b) - \mathcal{S}(\mathbf{v}; M_b)| = |\max_{k \in \mathcal{K}_{\text{InD}}} \mathbf{v}_k - \max_{k \in \mathcal{K}_{\text{InD}}} \mathbf{u}_k| \overset{(\star)}{\leq} \max_{k \in \mathcal{K}_{\text{InD}}} |\mathbf{v}_k - \mathbf{u}_k| = \|\mathbf{v} - \mathbf{u}\|_\infty$$

$$= \max_{k \in \mathcal{K}_{\text{InD}}} |\mathbf{u}_k - \mathbf{v}_k| \leq \sqrt{\sum_{k \in \mathcal{K}_{\text{InD}}} (\mathbf{u}_k - \mathbf{v}_k)^2} = \|\mathbf{u} - \mathbf{v}\|_2,$$

where we use the fact that the infinity norm of a vector is bounded above by its $l_2$-norm and the derivation of the $(\star)$ step is provided below. Without losing generality, we assume that $\max_{k \in \mathcal{K}_{\text{InD}}} \mathbf{v}_k \leq \max_{k \in \mathcal{K}_{\text{InD}}} \mathbf{u}_k$ (i.e. $\|\mathbf{v}\|_\infty \leq \|\mathbf{u}\|_\infty$) and let $k_{\mathbf{v}}$ denote the index of the largest value in $\mathbf{v}$. Then, we have the following:

$$|\max_{k \in \mathcal{K}_{\text{InD}}} \mathbf{v}_k - \max_{k \in \mathcal{K}_{\text{InD}}} \mathbf{u}_k| = \max_{k \in \mathcal{K}_{\text{InD}}} \mathbf{v}_k - \max_{k \in \mathcal{K}_{\text{InD}}} \mathbf{u}_k = \mathbf{v}k_{\mathbf{v}} - \max_{k \in \mathcal{K}_{\text{InD}}} \mathbf{u}_k$$

$$\leq \mathbf{v}_{k_{\mathbf{v}}} - \mathbf{u}_{k_{\mathbf{v}}} \leq \max_{k \in \mathcal{K}_{\text{InD}}} \mathbf{v}_k - \mathbf{u}_k \leq \max_{k \in \mathcal{K}_{\text{InD}}} |\mathbf{v}_k - \mathbf{u}_k|.$$

Note that in the other case where $\|\mathbf{v}\|_\infty \geq \|\mathbf{u}\|_\infty$, this can be easily derived using the same line of reasoning. Thus, under the Euclidean metric space, the Wasserstein score function $\mathcal{S}(f(\mathbf{x}); M_b)$ is shown to be 1-Lipschitz continuous. This completes the proof. $\qquad \square$

## B.3 DEFINITION OF *Neural Network Loss* & EXAMPLE

**Definition B.1.** *(Definition 1 restated) Let $\mathcal{F} : \mathbb{R}^d \longmapsto \mathbb{R}^K$ be a class of functions that projects the inputs to a $K$-dimensional probability vector, such that $f \in \mathcal{F}$ implies $\mathbf{1}_K(\cdot) - f \in \mathcal{F}$. Let $\Phi = \{\phi_1, \phi_2, \phi_3 : \mathbb{R}^K \longmapsto \mathbb{R}\}$ be a set of convex measuring functions that map a probability vector to a scalar score. Then, the neural network loss w.r.t. $\Phi$ among three distributions $p_1, p_2,$ and $p_3$ supported on $\mathbb{R}^d$ is defined as*

$$\mathcal{L}_{\mathcal{F},\Phi}(p_1, p_2, p_3) = \inf_{D \in \mathcal{F}} \mathbb{E}_{\mathbf{x} \sim p_1}[\phi_1(D(\mathbf{x}))] + \mathbb{E}_{\mathbf{x} \sim p_2}[\phi_2(D(\mathbf{x}))] + \mathbb{E}_{\mathbf{x} \sim p_3}[\phi_3(\mathbf{1}_K - D(\mathbf{x}))].$$

EXAMPLE 1. The *neural network loss* can be easily applied to our method by setting the three distributions $p_1$, $p_2$, and $p_3$ to $\mathbb{P}_{\text{InD}}$, $\mathbb{P}_{\text{OoD}}$, and $\mathbb{P}_{\text{G}}$, respectively. Unlike $\mathbb{P}_{\text{OoD}}$ and $\mathbb{P}_{\text{G}}$, $\mathbb{P}_{\text{InD}}$ is a joint distribution of $\mathbf{x}$ and $y$; however, we can simply decompose the marginal of $\mathbb{P}_{\text{InD}}$ into $K$ different classes according to its labels $y$, which results in a set of $K$ different distributions $\{\mathbb{P}_{\text{InD}}^y\}_{y=1}^K$ that are all supported on $\mathbb{R}^d$. Similarly, we replace the measuring function $\phi_1$ with a set of function $\{\phi_{1i}\}_{i=1}^K$, where $\phi_{1i}(\mathbf{x}) = -\log(\mathbf{x})[i]$. In other words, $p_1$ can be understood as the union of $\{\mathbb{P}_{\text{InD}}^y\}_{y=1}^K$, whereas the measuring function $\phi_1$ is conditioned on the distribution where $\mathbf{x}$ is drawn from. However, in either interpretation, this matches the first term in the definition seamlessly. According to the minimax objective function, we further identify that $\phi_2(\mathbf{x}) = \beta_{\text{OoD}}\|\mathbf{x}\|_\infty$ and $\phi_3(\mathbf{x}) = \beta_z\|\mathbf{x}\|_\infty$. Therefore, we have illustrated that with careful selection of the measuring functions, the *neural network loss* can recover the original objective function when $G$ is fixed.

EXAMPLE 2. Our proposed loss-measure, *neural network loss*, can be generalized to other settings by keeping the first term as detailed in Example 1 and setting $\phi_2$ and $\phi_3$ to be any score function that quantifies the predictive confidence of a discrete probability vector.

## B.4 PROOFS FOR THEOREM 1.

**Theorem B.1.** *(Theorem 1 restated) For a given discriminator $\bar{D}$, let $G_{\bar{D}}^\star$ be the optimal solution among all possible real-valued functions that map $\mathbb{R}^n$ to $\mathbb{R}^d$, then the Wasserstein scores of the generated data are lower bounded by the Wasserstein scores of OoD data, that is,*

$$\mathbb{E}_{\mathbf{z}\sim\mathbb{P}_{\mathbf{z}}}\left[\mathcal{S}(\bar{D}(G_{\bar{D}}^\star(\mathbf{z}));M_b)\right] \geq \mathbb{E}_{\mathbf{x}\sim\mathbb{P}_{OoD}(\mathbf{x})}\left[\mathcal{S}(\bar{D}(\mathbf{x});M_b)\right].$$

*Proof.* For a given discriminator $\bar{D}$, the optimization objective can be rewritten as the following:

$$\arg\max_G \mathbb{E}_{(\mathbf{x},y)\sim\mathbb{P}_{\text{InD}}(\mathbf{x},y)}[-\log(D(\mathbf{x})^\top \mathbf{e}_y)] - \beta_{\text{OoD}}\mathbb{E}_{\mathbf{x}\sim\mathbb{P}_{\text{OoD}}(\mathbf{x})}\mathcal{S}(D(\mathbf{x})) + \beta_z\mathbb{E}_{\mathbf{z}\sim\mathbb{P}_{\mathbf{z}}}\mathcal{S}(D(G(\mathbf{z})))$$

$$= \arg\max_G \beta_z\mathbb{E}_{\mathbf{z}\sim\mathbb{P}_{\mathbf{z}}}\mathcal{S}(D(G(\mathbf{z}))) + C.$$

The above problem aims at finding $G$ that generates synthetic data with the maximum possible Wasserstein score. Hence, assuming that the distribution of observed OoD samples can be learned by the generator, we directly get the desired result.

$\square$

## B.5 PROOFS FOR THEOREM 2.

**Theorem B.2.** *(Theorem 2. restated) Let $D$ and $G$ belong to the sets of all possible real-valued functions, in particular, neural networks, such that $D : \mathbb{R}^d \longmapsto \mathbb{R}^K$ and $G : \mathbb{R}^n \longmapsto \mathbb{R}^d$, respectively. Then, under optimality, $D^\star$ and $G^\star$ possess the following properties:*

$$D^\star(\mathbf{x}) = \begin{cases} \mathbf{e}_y & , (\mathbf{x},y)\sim\mathbb{P}_{InD}(\mathbf{x},y) \\ \frac{1}{K}\mathbf{1}_K & , \mathbf{x}\sim\mathbb{P}_{OoD}(\mathbf{x}) \end{cases} \text{ and } G^\star \in \{G : D^\star(G(\mathbf{z})) = \frac{1}{K}\mathbf{1}_K, \forall \mathbf{z}\sim\mathbb{P}_{\mathbf{z}}\}.$$

*Furthermore, if the discriminator $D$ is $\alpha$-Lipschitz continuous with respect to its inputs $\mathbf{x}$, where the Lipschitz constant $\alpha > 0$. Then, at optimality, $G^\star(\mathbf{z}) \not\sim \mathbb{P}_{InD}(\mathbf{x})$, $\forall \mathbf{z} \in \mathbb{P}_{\mathbf{z}}$; that is, the probability that the generated samples are In-Distribution is zero.*

*Proof.* We begin the proof by deriving the optimal solutions $D^\star$ and $G^\star$ in this minimax game. To find the optimal solution, we first evaluate the optima for $G$ given $D^\star$. For given $D^\star$, by Remark 1, we obtain

$$\arg\max_G \beta_z\mathbb{E}_{\mathbf{z}\sim\mathbb{P}_{\mathbf{z}}}\mathcal{S}(D^\star(G(\mathbf{z}))) = \arg\max_G \beta_z\mathbb{E}_{\mathbf{z}\sim\mathbb{P}_{\mathbf{z}}}\|(D^\star(G(\mathbf{z})))\|_\infty.$$

Knowing that the infinity norm of a probability vector is lower bounded by $\frac{1}{K}$, we get $G^\star = \{G : D^\star(G(\mathbf{z})) = \frac{1}{K}\mathbf{1}_K\}$. Now, let us focus on the outer minimization problem and find the optimal states for the discriminator $D^\star$. Let $f_{\text{InD}} : \mathbb{R}^d \longmapsto \mathbb{R}$ and $f_{\text{OoD}} : \mathbb{R}^d \longmapsto \mathbb{R}$ denote the marginal probability density functions of $\mathbb{P}_{\text{InD}}$ and $\mathbb{P}_{\text{OoD}}$, respectively. Correspondingly, let $\mathcal{X}_{\text{InD}} = \{\mathbf{x} :$

$f_{\text{InD}}(\mathbf{x}) > 0\}$ and $\mathcal{X}_{\text{OoD}} = \{\mathbf{x} : f_{\text{OoD}}(\mathbf{x}) > 0\}$ denote the set of all possible points with positive density in the two distributions. Using our previous arguments on $G^\star$ and Remark 1, we obtain

$$\arg\min_D \mathbb{E}_{(\mathbf{x},y)\sim\mathbb{P}_{\text{InD}}(\mathbf{x},y)}[-\log(D(\mathbf{x})^\top \mathbf{e}_y)] - \beta_{\text{OoD}}\mathbb{E}_{\mathbf{x}\sim\mathbb{P}_{\text{OoD}}(\mathbf{x})}[\mathcal{S}(D(\mathbf{x}); M_b)] + (1 - \frac{1}{K})$$

$$= \arg\min_D \mathbb{E}_{(\mathbf{x},y)\sim\mathbb{P}_{\text{InD}}(\mathbf{x},y)}[-\log(D(\mathbf{x})^\top \mathbf{e}_y)] - \beta_{\text{OoD}}\mathbb{E}_{\mathbf{x}\sim\mathbb{P}_{\text{OoD}}(\mathbf{x})}[1 - \|D(\mathbf{x})\|_\infty]$$

$$= \arg\min_D \mathbb{E}_{(\mathbf{x},y)\sim\mathbb{P}_{\text{InD}}(\mathbf{x},y)}[-\log(D(\mathbf{x})^\top \mathbf{e}_y)] + \beta_{\text{OoD}}\mathbb{E}_{\mathbf{x}\sim\mathbb{P}_{\text{OoD}}(\mathbf{x})}\|D(\mathbf{x})\|_\infty$$

$$= \arg\min_D \int_{\substack{(\mathbf{x},y)\sim\mathbb{P}_{\text{InD}}(\mathbf{x},y) \\ \mathbf{x}\in\mathcal{X}_{\text{InD}}\backslash\mathcal{X}_{\text{OoD}}}} -\log(D(\mathbf{x})^\top \mathbf{e}_y)\, dyd\mathbf{x} + \beta_{\text{OoD}}\int_{\mathbf{x}\in\mathcal{X}_{\text{OoD}}\backslash\mathcal{X}_{\text{InD}}} \|D(\mathbf{x})\|_\infty\, d\mathbf{x}$$

$$+ \underbrace{\int_{\substack{(\mathbf{x},y)\sim\mathbb{P}_{\text{InD}}(\mathbf{x},y) \\ \mathbf{x}\in\mathcal{X}_{\text{InD}}\cap\mathcal{X}_{\text{OoD}}}} -\log(D(\mathbf{x})^\top \mathbf{e}_y) + \beta_{\text{OoD}}\|D^\star(\mathbf{x})\|_\infty\, dyd\mathbf{x}}_{\text{Overlapping sets between }\mathcal{X}_{\text{InD}}\text{ and }\mathcal{X}_{\text{OoD}}\ (\triangle)}.$$

However, it is reasonable to assume that $\mathcal{X}_{\text{InD}} \cap \mathcal{X}_{\text{OoD}}$ is a zero-measure set as alluded in previous works (Liang et al., 2020; Du et al., 2022). If a sample belongs to both distributions, then it is meaningless to distinguish whether it is from InD or OoD, which also breaks the definition of OoD (Yang et al., 2021) and trivializes the OoD detection task. In addition, we observe that when the classification performance is not notoriously terrible, which is a valid assumption as the intersection of InD and OoD is assumed to be a zero-measure set, the integrand of $(\triangle)$ is bounded. Thus, the term $(\triangle)$ vanishes and the optimal value can be attained when $D^\star$ satisfies the following conditions:

$$D^\star(\mathbf{x}) = \begin{cases} \mathbf{e}_y, & (\mathbf{x}, y) \sim \mathbb{P}_{\text{InD}}(\mathbf{x}, y) \\ \frac{1}{K}\mathbf{1}_K, & \mathbf{x} \sim \mathbb{P}_{\text{OoD}}(\mathbf{x}) \\ \frac{1}{K}\mathbf{1}_K, & \mathbf{x} = G^\star(\mathbf{z}), \forall \mathbf{z} \sim \mathbb{P}_{\mathbf{z}} \end{cases}.$$

Note that the third condition has nothing to do with the outer minimization problem but it must be true under optimality. It is noteworthy that unlike classic GAN formulations (Goodfellow et al., 2020) where a unique optimal solution exists, in this minimax game, the best responses of $D$ and $G$ are actually sets; this is intuitive because the Wasserstein score calculation can be thought of as a lossy compression of information, rendering the optimal solutions to be possibly not unique.

Now, we will show that at optimality, the generated data $G^\star(\mathbf{z})$ will *never* fall into the In-Distribution, which is equivalent to showing that $f_{\text{InD}}(G^\star(\mathbf{z})) = 0,\ \forall \mathbf{z} \sim \mathbb{P}_{\mathbf{z}}$. Now suppose that $\mathbf{x}_g = G^\star(\mathbf{z}_0)$ for $\mathbf{z}_0 \sim \mathbb{P}_{\mathbf{z}}$ and $\mathbf{x}_g \sim \mathbb{P}_{\text{InD}}$, then the condition $\mathbf{x}_g \in \mathcal{X}_{\text{InD}}\ (\star)$ must hold. However, by Lipschitz continuity assumption of the discriminator $D^\star$ and Remark B.2., the following must hold for arbitrary points $\mathbf{x} \in \mathcal{X}_{\text{InD}}$:

$$\|\mathcal{S}(D^\star(\mathbf{x}_g)) - \mathcal{S}(D^\star(\mathbf{x}))\| \leq \|D^\star(\mathbf{x}_g) - D^\star(\mathbf{x})\|_2 \leq \alpha\|\mathbf{x}_g - \mathbf{x}\|_2.$$

Now plug in the optimal $D^\star$ of the discriminator, and we have that

$$\|\mathbf{x}_g - \mathbf{x}\|_2 \geq \frac{1}{\alpha}\left(1 - \frac{1}{K}\right) > 0,$$

which demonstrates that such $\mathbf{x} \neq \mathbf{x}_g,\ \forall \mathbf{x} \in \mathcal{X}_{\text{InD}}$. This contradicts with our assumption $(\star)$ that $\mathbf{x}_g \in \mathcal{X}_{\text{InD}}$, implying that $f_{\text{InD}}(G^\star(\mathbf{z})) = 0, \forall \mathbf{z} \sim \mathbb{P}_{\mathbf{z}}$, which is an equivalent statement of $G^\star(\mathbf{z}) \not\sim \mathbb{P}_{\text{InD}}, \forall \mathbf{z} \sim \mathbb{P}_{\mathbf{z}}$. This completes the proof. $\qquad\square$

### B.6 Proofs for Theorem 3.

**Theorem B.3.** (Theorem 3. restated) *Let $p_1, p_2$, and $p_3$ be three distributions and $\widehat{p}_1, \widehat{p}_2$, and $\widehat{p}_3$ be the empirical versions with at least $m$ samples each. Suppose the measuring functions $\phi_i \in \Phi$ are $L_{\phi_i}$-Lipschitz continuous and take values in $[l_i, u_i]$ for $i \in \{1, 2, 3\}$. Let the discriminator $D$ be $L$-Lipschitz continuous with respect to its parameter $\theta_D$ whose dimension is denoted by $p$. Then, there exists a universal constant $C$ such that when the empirical sample size $m \geq \max_i\{\frac{Cp(u_i-l_i)^2\log(LL_{\phi_i}p/\epsilon)}{\epsilon^2}\}$, we have with probability at least $1 - exp(-p)$ over the randomness of $\widehat{p}_1, \widehat{p}_2$, and $\widehat{p}_3$,*

$$|\mathcal{L}_{\mathcal{F},\Phi}(p_1, p_2, p_3) - \mathcal{L}_{\mathcal{F},\Phi}(\widehat{p}_1, \widehat{p}_2, \widehat{p}_3)| \leq \epsilon.$$

*Proof.* We prove the result using Chernoff bound. Here we slightly abuse the notation by omitting the subscript of parameter $\theta_D$ when representing the discriminator; that is, we use $D_\theta$ to denote the discriminator $D$ where $\theta_D$ is its parameter that is bounded in a $p$-dimensional unit ball. We aim to show that with high probability, for every possible discriminator $D_\theta$,

$$\left| \mathbb{E}_{\mathbf{x} \in p_1} [\phi_1(D_\theta(\mathbf{x}))] - \mathbb{E}_{\mathbf{x} \in \widehat{p}_1} [\phi_1(D_\theta(\mathbf{x}))] \right| \leq \epsilon/3, \tag{I-1}$$

$$\left| \mathbb{E}_{\mathbf{x} \in p_2} [\phi_2(D_\theta(\mathbf{x}))] - \mathbb{E}_{\mathbf{x} \in \widehat{p}_2} [\phi_2(D_\theta(\mathbf{x}))] \right| \leq \epsilon/3, \tag{I-2}$$

$$\text{and} \left| \mathbb{E}_{\mathbf{x} \in p_3} [\phi_3(\mathbf{1}_K - D_\theta(\mathbf{x}))] - \mathbb{E}_{\mathbf{x} \in \widehat{p}_3} [\phi_3(\mathbf{1}_K - D_\theta(\mathbf{x}))] \right| \leq \epsilon/3. \tag{I-3}$$

Then, with the above statements to be true, for optimal discriminator $D_\theta^\star$, we obtain

$$\mathcal{L}_{\mathcal{F},\Phi}(\widehat{p}_1, \widehat{p}_2, \widehat{p}_3) = \mathbb{E}_{\mathbf{x} \sim \widehat{p}_1} [\phi_1(D_\theta^\star(\mathbf{x}))] + \mathbb{E}_{\mathbf{x} \sim \widehat{p}_2} [\phi_2(D_\theta^\star(\mathbf{x}))] + \mathbb{E}_{\mathbf{x} \sim \widehat{p}_3} [\phi_2(\mathbf{1}_K - D_\theta^\star(\mathbf{x}))]$$

$$\leq \mathbb{E}_{\mathbf{x} \sim p_1} [\phi_1(D_\theta^\star(\mathbf{x}))] + \mathbb{E}_{\mathbf{x} \sim p_2} [\phi_2(D_\theta^\star(\mathbf{x}))] + \mathbb{E}_{\mathbf{x} \sim p_3} [\phi_2(\mathbf{1}_K - D_\theta^\star(\mathbf{x}))]$$

$$+ \left| \mathbb{E}_{\mathbf{x} \in p_1} [\phi_1(D_\theta^\star(\mathbf{x}))] - \mathbb{E}_{\mathbf{x} \in \widehat{p}_1} [\phi_1(D_\theta^\star(\mathbf{x}))] \right|$$

$$+ \left| \mathbb{E}_{\mathbf{x} \in p_2} [\phi_2(D_\theta^\star(\mathbf{x}))] - \mathbb{E}_{\mathbf{x} \in \widehat{p}_2} [\phi_2(D_\theta^\star(\mathbf{x}))] \right|$$

$$+ \left| \mathbb{E}_{\mathbf{x} \in p_3} [\phi_2(\mathbf{1}_K - D_\theta^\star(\mathbf{x}))] - \mathbb{E}_{\mathbf{x} \in \widehat{p}_3} [\phi_2(\mathbf{1}_K - D_\theta^\star(\mathbf{x}))] \right|$$

$$\leq \mathcal{L}_{\mathcal{F},\Phi}(p_1, p_2, p_3) + \epsilon.$$

The other direction is similar. Now it suffices to show that the claimed bounds (I-1), (I-2), and (I-3) are correct. We prove (I-1) as an example (proof of the other two is identical). Let $\mathcal{X}$ be a finite set such that every point in the parameter space $\theta_D \in \Theta_D$ is within distance $\epsilon/12LL_{\phi_1}$ of a point in $\mathcal{X}$. Standard construction of such $\epsilon/12LL_{\phi_1}$–net yields an $\mathcal{X}$ that satisfies $\log |\mathcal{X}| \leq O(p \log(LL_{\phi_1} p/\epsilon))$ (Haussler & Welzl, 1986). Therefore, for all $\theta_D \in \mathcal{X}$, by Chernoff bound, we can have that

$$P\left[ \left| \mathbb{E}_{\mathbf{x} \in p_1} [\phi_1(D_\theta(\mathbf{x}))] - \mathbb{E}_{\mathbf{x} \in \widehat{p}_1} [\phi_1(D_\theta(\mathbf{x}))] \right| \geq \frac{\epsilon}{6} \right] \leq 2e^{-\frac{\epsilon^2 m}{18(u_1 - l_1)^2}}.$$

Therefore, when $m \geq \frac{Cp(u_1 - l_1)^2 \log(LL_{\phi_1} p/\epsilon)}{\epsilon^2}$ for a sufficiently large constant $C$, we can union bound over all $\theta_D \in \mathcal{X}$ that with at least $1 - exp(-p)$ probability, for all $\theta_D \in \mathcal{X}$ we have that

$$\left| \mathbb{E}_{\mathbf{x} \in p_1} [\phi_1(D_\theta(\mathbf{x}))] - \mathbb{E}_{\mathbf{x} \in \widehat{p}_1} [\phi_1(D_\theta(\mathbf{x}))] \right| \leq \frac{\epsilon}{6}.$$

By the construction of $\mathcal{X}$ and the Lipschitz continuity assumption, we know that for every $\theta_D \in \Theta_D$, we can always find a $\theta_D{}' \in \mathcal{X}$, such that $\|\theta_D - \theta_D{}'\| \leq \epsilon/12LL_{\phi_1}$. Then, it follows that

$$\left| \mathbb{E}_{\mathbf{x} \in p_1} [\phi_1(D_\theta(\mathbf{x}))] - \mathbb{E}_{\mathbf{x} \in \widehat{p}_1} [\phi_1(D_\theta(\mathbf{x}))] \right| \leq \left| \mathbb{E}_{\mathbf{x} \in p_1} [\phi_1(D_{\theta'}(\mathbf{x}))] - \mathbb{E}_{\mathbf{x} \in \widehat{p}_1} [\phi_1(D_{\theta'}(\mathbf{x}))] \right|$$

$$+ \left| \mathbb{E}_{\mathbf{x} \in p_1} [\phi_1(D_\theta(\mathbf{x}))] - \mathbb{E}_{\mathbf{x} \in p_1} [\phi_1(D_{\theta'}(\mathbf{x}))] \right|$$

$$+ \left| \mathbb{E}_{\mathbf{x} \in \widehat{p}_1} [\phi_1(D_\theta(\mathbf{x}))] - \mathbb{E}_{\mathbf{x} \in \widehat{p}_1} [\phi_1(D_{\theta'}(\mathbf{x}))] \right|$$

$$\leq \epsilon/6 + \epsilon/12 + \epsilon/12$$

$$\leq \epsilon/3.$$

This completes the proof for (I-1) and the proofs for (I-2) and (I-3) follow from the same line of reasoning. The only difference is that the constant may be changed correspondingly when using different measuring functions. Therefore, when combining these three proofs together, it suffices to find their intersection, which provides us with the condition shown in the theorem that $m \geq \max_i \left\{ \frac{Cp(u_i - l_i)^2 \log(LL_{\phi_i} p/\epsilon)}{\epsilon^2} \right\}$ for sufficiently large $C$. This completes the proof. $\qquad \square$

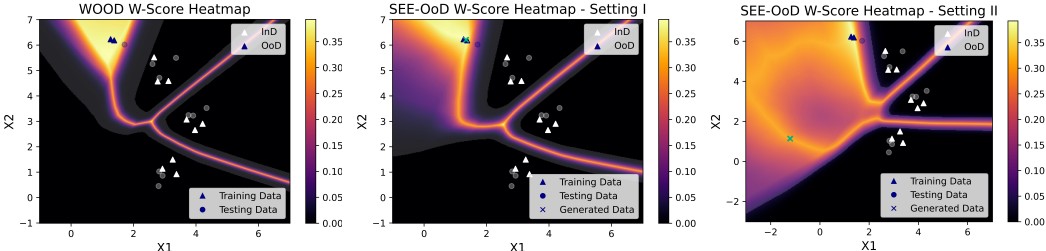

Figure C.1: (Figure 1 revisited) A 2D numerical illustration of the intuition behind SEE-OoD. In this figure, we present the Wasserstein score heatmaps of the WOOD (Wang et al., 2021) detector and *two* possible solution states of SEE-OoD after training, where brighter colors represent higher Wasserstein scores (i.e. OoD) and the shaded boundary is the InD/OoD decision boundary under 95% TNR.

### B.7 Proofs for Corollary 1.

**Corollary B.2.** *(Corollary 1 restated) In the setting of Theorem 3., suppose that $\{G^{(i)}\}_{i=0}^{N}$ be the $N$ generators in the $N$–iterations of the training, and assume $\log N \leq p$ and $\log N \ll d$. Then, there exists a universal constant $C$ such that when $m \geq \max_i \{\frac{Cp(u_i - l_i)^2 \log(LL_{\phi_i} p/\epsilon)}{\epsilon^2}\}$, with probability at least $1 - exp(-p)$, for all $t \in [N]$,*

$$|\mathcal{L}_{\mathcal{F},\Phi}(\mathbb{P}_{InD}, \mathbb{P}_{OoD}, \mathbb{P}_{G^{(t)}}) - \mathcal{L}_{\mathcal{F},\Phi}(\widehat{\mathbb{P}}_{InD}, \widehat{\mathbb{P}}_{OoD}, \widehat{\mathbb{P}}_{G^{(t)}})| \leq \epsilon.$$

*Proof.* The proof to this corollary is trivial as it follows from the proof of Theorem 3. The only difference here is that the generator distribution changes as the training goes on. However, we have fresh samples for every generator distribution $\widehat{\mathbb{P}}_{G^{(t)}}$ so this does not change our proof. □

## C NUMERICAL ILLUSTRATION

**DATA GENERATION PROCESS.** In this numerical simulation, both InD and OoD data are drawn from bivariate normal distributions with a diagonal covariance matrix. Respectively, three clusters of InD data are drawn from $\mathcal{N}(\boldsymbol{\mu}_i, \boldsymbol{\Sigma}_i)$ with $\boldsymbol{\mu}_1 = [4, 3]^\top$, $\boldsymbol{\mu}_2 = [3, 5]^\top$, and $\boldsymbol{\mu}_3 = [3, 1]^\top$, and $\boldsymbol{\Sigma}_i = 0.3^2 \mathbf{I}_2$, $\forall i = 1, 2, 3$. The OoD data are drawn from $\mathcal{N}(\boldsymbol{\mu}_{\text{OoD}}, \boldsymbol{\Sigma}_{\text{OoD}})$ where $\boldsymbol{\mu}_{\text{OoD}} = [1.5, 6]^\top$ and $\boldsymbol{\Sigma}_{\text{OoD}} = 0.3^2 \mathbf{I}_2$. For each cluster of InD and OoD data, 1000 training and 1000 testing data points are generated for this simulation study. However, to emulate scarce sample settings, only two OoD training samples are provided during the training stage, while all training samples for InD data are included. During the inference phase, all testing InD and OoD samples are presented to the trained detector.

**TRAINING CONFIGURATION.** As for classifier/discriminator architecture, a two-layer fully connected neural network with ReLU activation and a hidden dimension of 128 is used for both our method and the WOOD (Wang et al., 2021) baseline method. The generator architecture is symmetric to that of the discriminator but has an input dimension of $n = 2$ for the noise vector. The hyperparameters $(\beta_{\text{OoD}}, \beta_z, n_d, n_g, \eta_d, \eta_g)$ of the proposed SEE-OoD detector are set to $(1, 0.001, 2, 1, 0.0001, 0.0001)$ and $(1, 100, 1, 3, 0.0001, 0.001)$ in setting I (center plot) and setting II (right plot), respectively. As for WOOD, $\beta$ is set to 1 and the learning rate is set to 0.001. Both methods are trained with an Adam (Kingma & Ba, 2014) optimizer with $\beta_1 = 0.5$, $\beta_2 = 0.999$, and no learning rate decay.

**ANALYSIS OF RESULTS.** In the figure, the Wasserstein score of each point assigned by the detector is color-coded, where a brighter color represents a higher score. Intuitively, for an ideal OoD detector, we would expect to see darker regions around InD samples (i.e. low Wasserstein scores) while brighter regions in all other places (i.e. high Wasserstein scores). In this example, both methods achieve perfect InD/OoD separation (i.e. 100% TPR) given the complexity of bivariate normal variables is relatively low. The resulting OoD regions under 95% TNR on the test samples for both methods are indicated by the shaded areas in the figure. However, in both scenarios, our method provides a larger OoD rejection region with higher confidence and a tighter decision boundary. This

sheds light on the effectiveness of the generative adversarial training that we proposed and provides a straightforward illustration of the intuitions behind our method.

# D    EXPERIMENTAL SETUP

## D.1    MODEL ARCHITECTURE

**DISCRIMINATOR.** In this paper, we adopt the state-of-the-art classification model DenseNet (Huang et al., 2016) as the backbone for the discriminator, and specifically, we follow the configuration mentioned in (Wang et al., 2021; Huang et al., 2016) to use it with bottleneck blocks and set the depth, growth rate, and dropout rate to 100, 12, and 0, respectively. One difference is that we set the window size of the final average pooling layer from 8 to 7 so that the model is compatible with both 28-pixel and 32-pixel images without the need for cropping. As for other details of the architecture, we refer the audience to the original DenseNet paper (Huang et al., 2016).

**GENERATOR.** We design a CNN-based generator using combinations of transposed convolution layers, batch normalization layers, and pooling layers, where we use DC-GANs (Radford et al., 2015) as a reference. In our case, we design our generator to be more complex than that used by Lee et al. (2017) to make sure its capacity is sufficient for generating complicated images. Detailed configuration of the generator architecture and the output sizes of intermediate layers are provided in the following table. Note that the input to the generator is a batch of $B$ $n$-dimensional noise vectors that are drawn from some known prior distribution $\mathbb{P}_{\mathbf{z}}$.

Table 3: Detailed generator architecture. In the table below, $B$ denotes the batch size, $n$ is the input noise dimension, and the transposed convolutional block is represented in the format: Transposed Conv2D-[*kernel size*]-[*stride*]-[*padding*]. In addition, the pooling layer is represented as: AvgPooling2D-[*window size*]. Note that the architecture may be different depending on the dimension of the generated targets. The output size and configuration of each block for 3-channel images (i.e. SVHN & CIFAR-10) are provided, whereas those for grayscale images (i.e. MNIST & FashionMNIST) are presented in the parentheses.

| Layers / Blocks | Output Size | CNN Configuration / Description |
|---|---|---|
| Transformation | $B \times n \times 1 \times 1$ | Reshape the input noise vectors |
| CNN Block 1 | $B \times 512 \times 4 \times 4$ | Transposed Conv2D-4-1-0 + ReLU + BatchNorm |
| CNN Block 2 | $B \times 256 \times 8 \times 8$ | Transposed Conv2D-4-2-1 + ReLU + BatchNorm |
| CNN Block 3 | $B \times 128 \times 16(15) \times 16(15)$ | Transposed Conv2D-4(3)-2-1 + ReLU + BatchNorm |
| CNN Block 4 | $B \times 64 \times 32(29) \times 32(29)$ | Transposed Conv2D-4(3)-2-1 + ReLU + BatchNorm |
| CNN Block 5 | $B \times 3(1) \times 64(57) \times 64(57)$ | Transposed Conv2D-4(3)-2-1 + ReLU + BatchNorm |
| Pooling Layer | $B \times 3(1) \times 32(28) \times 32(28)$ | AvgPooling2D-2 + tanh activation |

## D.2    DISCUSSION ON HYPERPARAMETER

In this section, we report the hyperparameters for both baseline methods and the proposed SEE-OoD detector. In general, as mentioned in Section 3, the hyperparameters of these baseline methods are either chosen based on systematic tunings or borrowed from the original papers, while in our methods, the hyperparameters are decided based on the intuitions drawn from the simulation examples because the parameter spaces are too large to perform systematic tuning like grid search effectively.

### D.2.1    BASELINES

**MAHA & ODIN.** For Maha and ODIN, we borrow the hyperparameter tuning procedure from the original papers (Lee et al., 2018; Liang et al., 2020). For both methods, the magnitude of perturbation noise is chosen from $\{0, 0.0005, 0.001, 0.0014, 0.002, 0.0024, 0.005, 0.01, 0.05, 0.1, 0.2\}$. For ODIN, the temperature scaling constant is chosen from $\{1, 10, 100, 1000\}$. The final hyperparam-

eter is chosen based on the validation experiment, which is performed on a separate validation set that consists of 1000 InD/OoD data pairs. We refer the audience to the original papers for details.

**WOOD.** For the WOOD (Wang et al., 2021) method, we stick to the hyperparameter that is used in the original paper as the dataset pairs and model architectures that are used in this paper are similar to those experimented with in WOOD. Specifically, we set $\beta = 0.1$, $B_{\text{InD}} = 50$, and $B_{\text{OoD}} = 10$, and the model is trained for 100 epochs using an Adam (Kingma & Ba, 2014) optimizer with a learning rate of 0.001, $\beta_1 = 0.9$, and $\beta_2 = 0.999$ for all experiments.

**VOS.** For VOS (Du et al., 2022), we borrow the hyperparameters from the original paper. When sampling from the tails (i.e. $\epsilon$-likelihood region) of the class-conditional Gaussian distributions, the likelihood value $\epsilon$ is chosen based on the smallest likelihood in a pool of 10K samples. Furthermore, the InD queue size is set to 1K. In addition, the loss weight $\beta$ is set to 0.1, and the regularization on virtual outliers is introduced at epoch 40. In terms of optimization, the model is trained for 100 epochs using an SGD optimizer with a learning rate of 0.1, Nesterov momentum $\mu = 0.9$, and weight decay $\lambda = 0.0001$. For more details regarding the meanings and intuitions behind these hyperparameters, we refer the audience to Appendix C of the original paper.

**GAN-SYNTHESIS.** To train a classifier with joint confidence loss (Lee et al., 2017), which we refer to as the *GAN-synthesis* method in this paper, we select the regularization weight in the second term of the joint confidence loss (i.e. KL divergence regularization term) from the set $\{0.001, 0.01, 0.1, 1, 2\}$. The classifier is trained using an SGD optimizer with a learning rate of 0.1, Nesterov momentum $\mu = 0.9$, and weight decay $\lambda = 0.0001$ for 100 epochs, where the learning rate decays by a factor of 0.1 at the 50th and 75th epoch and the training and validation batch size are both set to 128. In addition, Lee et al. (2017) uses an auxiliary GAN (Goodfellow et al., 2020) to synthesize virtual outliers. We stick to the GAN architecture that is introduced in the original paper, and both the discriminator and the generator are optimized with an Adam (Kingma & Ba, 2014) optimizer with a learning rate of 0.001, $\beta_1 = 0.5$ and $\beta_2 = 0.999$.

**ENERGY & ENERGY + FINETUNING & OE.** According to Liu et al. (2020), the energy score can be used in a parameter-free manner by simply setting $T = 1$. As for the Energy + Finetuning (EFT) method, we finetune the pre-trained model using an SGD optimizer with a learning rate of 0.001, Nesterov momentum $\mu = 0.9$, and weight decay $\lambda = 0.0005$ for 100 epochs, where the training and validation batch size are both set to 128 and the margin parameters are $m_{\text{in}} = -25$ and $m_{\text{out}} = -7$. For OE, we also follow the experimental setup introduced in Liu et al. (2020).

In addition, DenseNet (Huang et al., 2016) with the same configuration as introduced in Appendix D.1 is used as the backbone classifier for all baseline methods for fair comparison. For methods that require pre-training, the classification model is trained for 300 epochs with an SGD optimizer with a learning rate of 0.1, Nesterov momentum $\mu = 0.9$, and weight decay $\lambda = 0.0001$. The learning rate is scheduled to decrease by 0.1 at the 150th and 225th epochs.

### D.2.2 SEE-OOD

In this section, we present the parameters of interest of the proposed SEE-OoD detector.

**METHOD PARAMETERS.** We set $\beta_{\text{OoD}} = 0.1$, $\beta_z = 0.001$, $n_d = n_g = 1$, and $n = 96$ to balance the loss between each term in the objective function in the training process. We have found that setting $n_d > 1$ or $n_g > 1$ may slightly improve performances in some scenarios; however, we stick to $n_d = n_g = 1$ for all experiments because the multistep update scheme increases the training cost significantly whereas the performance gain is usually negligible. Note that real dataset experiments are very different from the aforementioned 2D numerical simulation due to the complexity of the dataset; hence $\beta_{\text{OoD}} > \beta_z$ does not necessarily imply a dominant discriminator after training. In fact, for real dataset experiments, it is difficult to precisely control the underlying dynamics between the generator and the discriminator by manipulating the hyperparameters. However, one huge advantage of our method over others is that the SEE-OoD detector is guaranteed to explore OoD spaces and improve OoD detection performance regardless of the equilibrium states it finally achieves.

**OPTIMIZATION PARAMETERS.** Both the discriminator and the generator of the proposed SEE-OoD are trained with an Adam (Kingma & Ba, 2014) optimizer with $\beta_1 = 0.5$ and $\beta_2 = 0.999$. Details about learning rates $\eta_d, \eta_g$ are presented in Table 4 below. Typically, batch sizes are set to $B_{\text{InD}} = 64$, $B_{\text{OoD}} = 32$, and $B_G = 64$. In our experiments, for scenarios where the observed OoD

sample size (i.e. $N_{\text{OoD}}$) is small, we use $B_{\text{InD}} = 50$, $B_{\text{OoD}} = 10$, and $B_G = 50$ to impose more stochasticity in the iterative training process. Note that when the OoD batch size is greater than the number of observed OoD samples, we simply take $B_{\text{OoD}} = \min\{B_{\text{OoD}}, N_{\text{OoD}}\}$, where the quantity $N_{\text{OoD}} = \#$ of observed sample per class $\times \#$ of observed classes. This convention enables us to present the hyperparameters in a more elegant way without taking into account $N_{\text{OoD}}$. For simplicity purposes, we denote these two different batch size combinations by $B_n$ and $B_s$, representing **n**ormal and **s**carce sample cases, respectively. More details about batch sizes can also be found in Table 4.

Table 4: SEE-OoD optimization parameters. In this table, all of the optimization parameters of interest are presented in a tuple form (Batch size, $\eta_d$, $\eta_g$, $N_{\text{epochs}}$) for simplicity. Note that $N_{\text{OoD}} = \#$ of observed sample per class $\times \#$ of observed classes, where $\#$ of observed classes varies based on different experiment datasets and regimes while $\#$ of observed sample per class is controlled and manipulated in all experiments.

| Experiment Dataset | Regime | Optimization Parameters (Batch size, $\eta_d$, $\eta_g$, $N_{\text{epochs}}$) |
|---|---|---|
| MNIST | I | $(B_s, 0.001, 0.005, 16)$ |
| FashionMNIST | I | $(B_n, 0.001, 0.001, 16)$ |
| | II | $(B_n, 0.001, 0.001, 16)$ |
| SVHN | I | $(B_n, 0.001, 0.001, 16)$ if $N_{\text{OoD}} \geq 32$; otherwise $(B_s, 0.001, 0.005, 16)$ |
| | II | $(B_n, 0.001, 0.001, 16)$ if $N_{\text{OoD}} \geq 32$; otherwise $(B_s, 0.001, 0.005, 16)$ |
| MNIST-FashionMNIST | I | $(B_n, 0.001, 0.001, 16)$ |
| CIFAR10-SVHN | I | $(B_n, 0.001, 0.001, 16)$ |

# E    EXPERIMENTAL RESULTS

In this section, we report all experimental results in tabular form. For *OoD-based methods* WOOD (Wang et al., 2021) and the proposed SEE-OoD, we also report the MAD of Monte Carlo repetitions. The TPRs under both $95\%$ and $99\%$ TNR thresholds and AUROC are reported for all methods. In convention, when reporting the results, (1) TPR or AUROC that is greater than $99.99\%$ is rounded to $100\%$, and (2) MAD that is lower than $0.01\%$ is reported as $< 0.01\%$. Furthermore, for Regime II experiments, we only report the results for methods that utilize OoD samples in the training stage. In terms of classification accuracy, all methods in all experiments achieve state-of-the-art performance after sufficient training, which is not surprising as DenseNet is known to perform well in image classification tasks. The tabular results can be found starting in next page.

One exception is about the *GAN-synthesis* method (Lee et al., 2017) in CIFAR10-SVHN *Between-Dataset* experiments, where the result is not applicable (i.e. N/A). For this method, we conducted systematic hyperparameter tunings based on the instructions provided in the original paper but this method fails to converge when we utilize DenseNet as the backbone for the classification model, which is also used for all other experiments. One conjecture on this phenomenon is that instead of tuning hyperparameters, one may also need to carefully balance the capacity and expressive power between the backbone classifier and the auxiliary GAN used to generate boundary samples; hence if one dominates over the other, it may be difficult to learn a classification model. Nonetheless, experimental results in other settings are promising enough to show the advantages of our methods over the baselines.

### E.1 REGIME I EXPERIMENTAL RESULTS

### E.1.1 MNIST WITHIN-DATASET EXPERIMENT

Table 5: TPR for MNIST *Within-Dataset* OoD detection

| TNR | Method | Number of observed OoD samples for **EACH** OoD class | | | | | | | | | | |
|---|---|---|---|---|---|---|---|---|---|---|---|---|
| | | 4 | 8 | 16 | 32 | 64 | 128 | 256 | 512 | 1024 | 2048 | 4096 |
| 95% | ODIN | | | | | | 98.88 | | | | | |
| | Maha | | | | | | 86.16 | | | | | |
| | Energy | | | | | | 99.10 | | | | | |
| | VOS | | | | | | 98.94 | | | | | |
| | GAN-Synthesis | | | | | | 99.30 | | | | | |
| | OE | 98.31 | 99.00 | 98.33 | 99.22 | 99.48 | 99.62 | 99.36 | 99.24 | 99.70 | 99.66 | 99.86 |
| | Energy + FT | 99.86 | 99.88 | 99.74 | 99.84 | 99.80 | 99.94 | 99.88 | 99.98 | 100.0 | 100.0 | 100.0 |
| | WOOD | 99.69 (<0.01) | 99.66 (<0.01) | 99.68 (<0.01) | 99.92 (<0.01) | 99.75 (<0.01) | 99.94 (<0.01) | 99.92 (<0.01) | **100.0** (0.0) | **100.0** (0.0) | **100.0** (0.0) | **100.0** (0.0) |
| | SEE-OoD | **100.0** (0.0) | **100.0** (0.0) | **100.0** (0.0) | **100.0** (0.0) | **100.0** (0.0) | **100.0** (0.0) | **100.0** (0.0) | 99.23 (1.03) | **100.0** (0.0) | **100.0** (0.0) | **100.0** (0.0) |
| 99% | ODIN | | | | | | 90.63 | | | | | |
| | Maha | | | | | | 60.43 | | | | | |
| | Energy | | | | | | 92.63 | | | | | |
| | VOS | | | | | | 87.56 | | | | | |
| | GAN-Synthesis | | | | | | 91.29 | | | | | |
| | OE | 94.82 | 96.47 | 94.94 | 96.77 | 97.21 | 98.47 | 98.55 | 99.04 | 99.70 | 99.66 | 99.86 |
| | Energy + FT | 97.29 | 97.93 | 97.39 | 98.63 | 98.59 | 99.24 | 99.28 | 99.70 | 99.82 | 99.90 | 99.96 |
| | WOOD | 98.26 (0.29) | 98.84 (0.22) | 98.89 (0.22) | 99.49 (<0.01) | 98.92 (0.12) | 99.66 (<0.01) | 99.65 (<0.01) | **99.85** (<0.01) | 99.98 (<0.01) | 99.98 (<0.01) | **100.0** (0.0) |
| | SEE-OoD | **100.0** (0.0) | **100.0** (0.0) | **100.0** (0.0) | **100.0** (0.0) | **99.97** (<0.01) | **100.0** (0.0) | **100.0** (0.0) | 99.11 (1.19) | **100.0** (0.0) | **100.0** (0.0) | **100.0** (0.0) |

Table 6: AUROC for MNIST *Within-Dataset* OoD detection

| Method | Number of training OoD samples for **EACH** OoD class | | | | | | | | | | |
|---|---|---|---|---|---|---|---|---|---|---|---|
| | 4 | 8 | 16 | 32 | 64 | 128 | 256 | 512 | 1024 | 2048 | 4096 |
| ODIN | | | | | | 99.49 | | | | | |
| Maha | | | | | | 96.19 | | | | | |
| Energy | | | | | | 99.48 | | | | | |
| VOS | | | | | | 99.36 | | | | | |
| GAN-Synthesis | | | | | | 99.52 | | | | | |
| OE | 98.94 | 99.29 | 98.92 | 99.45 | 99.59 | 99.69 | 99.60 | 99.52 | 99.76 | 99.79 | 99.90 |
| Energy + FT | 99.87 | 99.90 | 99.86 | 99.93 | 99.93 | 99.96 | 99.96 | **99.98** | 99.99 | 99.99 | **100.0** |
| WOOD | 99.73 (<0.01) | 99.75 (<0.01) | 99.80 (< 0.01) | 99.92 (<0.01) | 99.88 (<0.01) | 99.94 (<0.01) | 99.92 (<0.01) | 99.95 (<0.01) | 99.97 (<0.01) | 99.97 (<0.01) | 99.96 (<0.01) |
| SEE-OoD | **100.0** (0.0) | **100.0** (0.0) | **100.0** (0.0) | **100.0** (0.0) | **100.0** (<0.01) | **100.0** (0.0) | **100.0** (0.0) | 99.91 (0.01) | **100.0** (0.0) | **100.0** (0.0) | **100.0** (0.0) |

### E.1.2 FASHIONMNIST WITHIN-DATASET EXPERIMENT

Table 7: TPR for FashionMNIST *Within-Dataset* OoD detection

| TNR | Method | Number of training OoD samples for **EACH** OoD class | | | | | | | | | | |
|---|---|---|---|---|---|---|---|---|---|---|---|---|
| | | 4 | 8 | 16 | 32 | 64 | 128 | 256 | 512 | 1024 | 2048 | 4096 |
| 95% | **ODIN** | | | | | | 68.60 | | | | | |
| | **Maha** | | | | | | 55.90 | | | | | |
| | **Energy** | | | | | | 65.25 | | | | | |
| | **VOS** | | | | | | 59.54 | | | | | |
| | **GAN-Synthesis** | | | | | | 41.81 | | | | | |
| | **OE** | 43.10 | 44.71 | 53.50 | 60.89 | 67.29 | 73.85 | 72.14 | 76.59 | 78.50 | 95.74 | 89.56 |
| | **Energy + FT** | 75.34 | 84.80 | 90.99 | 93.95 | 95.55 | 97.03 | 96.33 | 96.90 | 97.29 | 97.84 | 98.58 |
| | **WOOD** | 72.33 (3.24) | 78.25 (0.97) | 88.52 (1.16) | 93.90 (0.90) | 95.88 (<0.01) | 96.78 (0.36) | 98.18 (0.24) | 98.60 (0.17) | 99.17 (<0.01) | 99.40 (0.13) | 99.47 (<0.01) |
| | **SEE-OoD** | **79.37** (3.06) | **99.85** (0.10) | **99.95** (<0.01) | **100.0** (0.0) | **100.0** (0.0) | **99.98** (<0.01) | **99.97** (<0.01) | **100.0** (0.0) | **100.0** (0.0) | **100.0** (0.0) | **100.0** (0.0) |
| 99% | **ODIN** | | | | | | 40.40 | | | | | |
| | **Maha** | | | | | | 38.60 | | | | | |
| | **Energy** | | | | | | 35.74 | | | | | |
| | **VOS** | | | | | | 32.12 | | | | | |
| | **GAN-Synthesis** | | | | | | 21.08 | | | | | |
| | **OE** | 12.79 | 12.65 | 19.69 | 23.90 | 28.67 | 34.27 | 33.75 | 39.52 | 40.80 | 49.57 | 55.02 |
| | **Energy + FT** | **56.96** | 64.96 | 76.54 | 84.71 | 87.65 | 91.59 | 90.99 | 93.03 | 94.88 | 96.20 | 96.88 |
| | **WOOD** | 49.43 (0.69) | 63.81 (1.81) | 81.00 (0.97) | 89.93 (1.09) | 92.82 (1.08) | 94.90 (0.37) | 96.10 (0.17) | 97.30 (<0.01) | 97.95 (0.17) | 98.67 (0.11) | 98.87 (0.11) |
| | **SEE-OoD** | 47.53 (1.91) | **98.33** (1.11) | **99.75** (0.33) | **99.98** (<0.01) | **100.0** (0.0) | **99.97** (<0.01) | **99.83** (0.22) | **100.0** (0.0) | **100.0** (0.0) | **100.0** (0.0) | **100.0** (0.0) |

Table 8: AUROC for FashionMNIST *Within-Dataset* OoD detection

| Method | Number of training OoD samples for **EACH** OoD class | | | | | | | | | | |
|---|---|---|---|---|---|---|---|---|---|---|---|
| | 4 | 8 | 16 | 32 | 64 | 128 | 256 | 512 | 1024 | 2048 | 4096 |
| **ODIN** | | | | | | 93.27 | | | | | |
| **Maha** | | | | | | 83.18 | | | | | |
| **Energy** | | | | | | 93.30 | | | | | |
| **VOS** | | | | | | 89.08 | | | | | |
| **GAN-Synthesis** | | | | | | 87.21 | | | | | |
| **OE** | 84.77 | 88.30 | 90.38 | 92.49 | 93.61 | 95.02 | 94.14 | 94.49 | 95.18 | 96.20 | 96.94 |
| **Energy + FT** | 94.72 | 96.61 | 97.98 | 98.64 | 99.00 | 99.35 | 98.96 | 99.08 | 99.29 | 99.44 | 99.60 |
| **WOOD** | 94.27 (1.32) | 95.67 (0.15) | 97.43 (0.16) | 98.45 (0.29) | 99.10 (<0.01) | 99.05 (<0.01) | 99.04 (0.26) | 99.27 (<0.01) | 99.24 (0.25) | 99.32 (0.22) | 99.18 (<0.01) |
| **SEE-OoD** | **97.21** (0.14) | **99.94** (<0.01) | **100.0** (0.0) | **100.0** (0.0) | **100.0** (0.0) | **100.0** (<0.01) | **100.0** (<0.01) | **100.0** (0.0) | **100.0** (0.0) | **100.0** (0.0) | **100.0** (0.0) |

### E.1.3 SVHN WITHIN-DATASET EXPERIMENT

Table 9: TPR for SVHN *Within-Dataset* OoD detection

| TNR | Method | Number of training OoD samples for **EACH** OoD class | | | | | | | | | | |
|---|---|---|---|---|---|---|---|---|---|---|---|---|
| | | 4 | 8 | 16 | 32 | 64 | 128 | 256 | 512 | 1024 | 2048 | 4096 |
| | **ODIN** | | | | | | 70.38 | | | | | |
| | **Maha** | | | | | | 21.55 | | | | | |
| | **Energy** | | | | | | 72.72 | | | | | |
| | **VOS** | | | | | | 65.93 | | | | | |
| 95% | **GAN-Synthesis** | | | | | | 34.41 | | | | | |
| | **OE** | 63.99 | 62.70 | 68.60 | 71.71 | 75.33 | 79.57 | 83.78 | 84.76 | 88.60 | 91.09 | 93.76 |
| | **Energy + FT** | **81.54** | 81.87 | 86.08 | 87.96 | 91.24 | 92.60 | 94.22 | 94.84 | 95.58 | 96.59 | 97.60 |
| | **WOOD** | 59.91 (0.79) | 69.20 (3.36) | 73.03 (0.31) | 83.95 (1.71) | 86.66 (1.98) | 90.33 (0.48) | 94.23 (0.54) | 95.55 (0.39) | 97.13 (0.21) | 97.48 (<0.01) | 98.10 (<0.01) |
| | **SEE-OoD** | 59.01 (6.55) | **95.45** (1.52) | **98.64** (0.85) | **99.61** (0.19) | **99.88** (0.12) | **99.76** (0.29) | **100.0** (0.0) | **100.0** (0.0) | **100.0** (0.0) | **100.0** (0.0) | **100.0** (0.0) |
| | **ODIN** | | | | | | 38.27 | | | | | |
| | **Maha** | | | | | | 5.10 | | | | | |
| | **Energy** | | | | | | 38.53 | | | | | |
| | **VOS** | | | | | | 25.59 | | | | | |
| 99% | **GAN-Synthesis** | | | | | | 9.97 | | | | | |
| | **OE** | 18.65 | 18.99 | 22.06 | 27.80 | 34.07 | 38.43 | 41.90 | 48.33 | 55.48 | 59.11 | 62.58 |
| | **Energy + FT** | **53.52** | 54.35 | 61.81 | 66.94 | 72.69 | 76.37 | 79.42 | 81.32 | 86.02 | 88.17 | 89.86 |
| | **WOOD** | 18.74 (0.76) | 38.99 (0.40) | 49.89 (3.22) | 67.19 (1.49) | 74.23 (3.30) | 81.48 (0.38) | 86.81 (1.13) | 90.37 (0.85) | 93.33 (0.47) | 94.22 (0.29) | 95.63 (<0.01) |
| | **SEE-OoD** | 18.41 (4.31) | **75.59** (3.43) | **90.96** (5.07) | **98.01** (0.72) | **99.50** (0.15) | **99.26** (0.72) | **99.92** (0.11) | **100.0** (0.0) | **100.0** (0.0) | **100.0** (0.0) | **100.0** (0.0) |

Table 10: AUROC for SVHN *Within-Dataset* OoD detection

| Method | Number of training OoD samples for **EACH** OoD class | | | | | | | | | | |
|---|---|---|---|---|---|---|---|---|---|---|---|
| | 4 | 8 | 16 | 32 | 64 | 128 | 256 | 512 | 1024 | 2048 | 4096 |
| **ODIN** | | | | | | 93.65 | | | | | |
| **Maha** | | | | | | 69.62 | | | | | |
| **Energy** | | | | | | 94.37 | | | | | |
| **VOS** | | | | | | 93.30 | | | | | |
| **GAN-Synthesis** | | | | | | 85.07 | | | | | |
| **OE** | 93.05 | 92.77 | 93.98 | 94.46 | 95.12 | 95.93 | 96.11 | 95.78 | 96.36 | 96.94 | 97.68 |
| **Energy + FT** | **96.47** | 96.58 | 97.41 | 97.79 | 98.22 | 98.54 | 98.71 | 98.75 | 98.95 | 99.08 | 99.29 |
| **WOOD** | 87.36 (0.37) | 89.68 (<0.01) | 93.66 (0.88) | 95.96 (0.77) | 97.34 (0.29) | 97.21 (0.12) | 98.62 (<0.01) | 98.57 (<0.01) | 99.09 (<0.01) | 99.05 (<0.01) | 99.08 (<0.01) |
| **SEE-OoD** | 93.14 (0.95) | **99.02** (0.10) | **99.66** (<0.01) | **99.91** (<0.01) | **99.96** (<0.01) | **99.95** (<0.01) | **100.0** (0.0) | **100.0** (0.0) | **100.0** (0.0) | **100.0** (0.0) | **100.0** (0.0) |

### E.1.4  MNIST-FASHIONMNIST BETWEEN-DATASET EXPERIMENT

Table 11: TPR for MNIST-FashionMNIST *Between-Dataset* OoD detection

| TNR | Method | \multicolumn{11}{c}{Number of training OoD samples for **EACH** OoD class} |
| | | 4 | 8 | 16 | 32 | 64 | 128 | 256 | 512 | 1024 | 2048 | 4096 |
|---|---|---|---|---|---|---|---|---|---|---|---|---|
| 95% | **ODIN** | | | | | | 94.68 | | | | | |
| | **Maha** | | | | | | **100.0** | | | | | |
| | **Energy** | | | | | | 77.72 | | | | | |
| | **VOS** | | | | | | 81.03 | | | | | |
| | **GAN-Synthesis** | | | | | | 96.61 | | | | | |
| | **OE** | 100.0 | 99.99 | 100.0 | 100.0 | 99.99 | 99.99 | 100.0 | 100.0 | 100.0 | 100.0 | 100.0 |
| | **Energy + FT** | 99.99 | 100.0 | 100.0 | 100.0 | 100.0 | 100.0 | 100.0 | 100.0 | 100.0 | 100.0 | 100.0 |
| | **WOOD** | 100.0 (0.0) | 100.0 (0.0) | 100.0 (0.0) | 100.0 (0.0) | 100.0 (0.0) | 100.0 (0.0) | 100.0 (0.0) | 100.0 (0.0) | 100.0 (0.0) | 100.0 (0.0) | 100.0 (0.0) |
| | **SEE-OoD** | 100.0 (0.0) | 100.0 (0.0) | 100.0 (0.0) | 100.0 (0.0) | 100.0 (0.0) | 100.0 (0.0) | 100.0 (0.0) | 100.0 (0.0) | 100.0 (0.0) | 100.0 (0.0) | 100.0 (0.0) |
| 99% | **ODIN** | | | | | | 66.61 | | | | | |
| | **Maha** | | | | | | **100.0** | | | | | |
| | **Energy** | | | | | | 35.98 | | | | | |
| | **VOS** | | | | | | 46.38 | | | | | |
| | **GAN-Synthesis** | | | | | | 76.09 | | | | | |
| | **OE** | 97.98 | 98.58 | 99.25 | 99.21 | 99.54 | 99.70 | 99.73 | 99.84 | 99.90 | 100.0 | 100.0 |
| | **Energy + FT** | 99.73 | 99.81 | 99.93 | 99.95 | 100.0 | 99.98 | 100.0 | 100.0 | 100.0 | 100.0 | 100.0 |
| | **WOOD** | 99.91 (<0.01) | 99.95 (<0.01) | 100.0 (0.0) | 100.0 (0.0) | 100.0 (0.0) | 100.0 (0.0) | 100.0 (0.0) | 100.0 (0.0) | 100.0 (0.0) | 100.0 (0.0) | 100.0 (0.0) |
| | **SEE-OoD** | 100.0 (0.0) | 100.0 (0.0) | 100.0 (0.0) | 100.0 (0.0) | 100.0 (0.0) | 100.0 (0.0) | 100.0 (0.0) | 100.0 (0.0) | 100.0 (0.0) | 100.0 (0.0) | 100.0 (0.0) |

Table 12: AUROC for MNIST-FashionMNIST *Between-Dataset* OoD detection

| Method | \multicolumn{11}{c}{Number of training OoD samples for **EACH** OoD class} |
| | 4 | 8 | 16 | 32 | 64 | 128 | 256 | 512 | 1024 | 2048 | 4096 |
|---|---|---|---|---|---|---|---|---|---|---|---|
| **ODIN** | | | | | | 98.05 | | | | | |
| **Maha** | | | | | | **100.0** | | | | | |
| **Energy** | | | | | | 91.42 | | | | | |
| **VOS** | | | | | | 95.31 | | | | | |
| **GAN-Synthesis** | | | | | | 98.17 | | | | | |
| **OE** | 99.86 | 99.89 | 99.92 | 99.94 | 99.95 | 99.96 | 99.96 | 99.98 | 99.99 | 99.99 | 99.99 |
| **Energy + FT** | 99.98 | 99.99 | 100.0 | 100.0 | 100.0 | 100.0 | 100.0 | 100.0 | 100.0 | 100.0 | 100.0 |
| **WOOD** | 100.0 (<0.01) | 100.0 (<0.01) | 100.0 (0.0) | 100.0 (0.0) | 100.0 (0.0) | 100.0 (0.0) | 100.0 (0.0) | 100.0 (0.0) | 100.0 (0.0) | 100.0 (0.0) | 100.0 (0.0) |
| **SEE-OoD** | 100.0 (<0.01) | 100.0 (<0.01) | 100.0 (0.0) | 100.0 (0.0) | 100.0 (0.0) | 100.0 (0.0) | 100.0 (0.0) | 100.0 (0.0) | 100.0 (0.0) | 100.0 (0.0) | 100.0 (0.0) |

### E.1.5 CIFAR10-SVHN BETWEEN-DATASET EXPERIMENT

Table 13: TPR for CIFAR10-SVHN *Between-Dataset* OoD detection

| TNR | Method | Number of training OoD samples for **EACH** OoD class | | | | | | | | | | |
|---|---|---|---|---|---|---|---|---|---|---|---|---|
| | | 4 | 8 | 16 | 32 | 64 | 128 | 256 | 512 | 1024 | 2048 | 4096 |
| 95% | **ODIN** | | | | | | 81.47 | | | | | |
| | **Maha** | | | | | | 88.13 | | | | | |
| | **Energy** | | | | | | 78.10 | | | | | |
| | **VOS** | | | | | | 80.84 | | | | | |
| | **GAN-Synthesis** | | | | | | N/A | | | | | |
| | **OE** | 87.94 | 98.50 | 90.23 | 84.64 | 89.05 | 92.47 | 90.76 | 93.66 | 95.94 | 96.71 | 97.25 |
| | **Energy + FT** | 98.67 | 99.01 | 99.33 | 99.64 | 99.54 | 99.79 | 99.83 | 99.92 | 99.97 | 99.96 | **100.0** |
| | **WOOD** | 95.82 (0.57) | 96.72 (0.31) | 97.53 (0.20) | 98.20 (0.01) | 99.00 (<0.01) | 98.95 (0.15) | 98.90 (0.01) | 99.17 (0.16) | 99.46 (0.13) | 99.51 (<0.01) | 99.56 (<0.01) |
| | **SEE-OoD** | **99.09** (0.51) | **99.78** (0.15) | **99.75** (0.16) | **100.0** (0.0) | **100.0** (0.0) | **99.92** (0.01) | **99.97** (<0.01) | **99.96** (<0.01) | **100.0** (0.0) | **100.0** (0.0) | **100.0** (0.0) |
| 99% | **ODIN** | | | | | | 61.15 | | | | | |
| | **Maha** | | | | | | 70.20 | | | | | |
| | **Energy** | | | | | | 43.90 | | | | | |
| | **VOS** | | | | | | 44.01 | | | | | |
| | **GAN-Synthesis** | | | | | | N/A | | | | | |
| | **OE** | 70.77 | 71.95 | 69.00 | 54.65 | 63.84 | 69.69 | 67.32 | 71.17 | 78.15 | 80.45 | 83.54 |
| | **Energy + FT** | 96.50 | 97.13 | 97.76 | 97.97 | 98.43 | 99.15 | 99.37 | 99.65 | 99.88 | 99.89 | **99.91** |
| | **WOOD** | 90.27 (2.33) | 94.35 (0.65) | 95.87 (0.17) | 97.38 (0.32) | 98.17 (<0.01) | 98.24 (0.14) | 98.12 (0.30) | 98.61 (0.18) | 99.05 (0.16) | 99.02 (0.17) | 99.15 (<0.01) |
| | **SEE-OoD** | **98.42** (0.60) | **99.51** (0.41) | **99.62** (0.25) | **99.98** (<0.01) | **100.0** (0.0) | **99.90** (0.01) | **99.96** (<0.01) | **99.81** (0.03) | **100.0** (0.0) | **100.0** (0.0) | **99.91** (<0.01) |

Table 14: AUROC for CIFAR10-SVHN *Between-Dataset* OoD detection

| Method | Number of training OoD samples for **EACH** OoD class | | | | | | | | | | |
|---|---|---|---|---|---|---|---|---|---|---|---|
| | 4 | 8 | 16 | 32 | 64 | 128 | 256 | 512 | 1024 | 2048 | 4096 |
| **ODIN** | | | | | | 95.04 | | | | | |
| **Maha** | | | | | | 95.72 | | | | | |
| **Energy** | | | | | | 91.66 | | | | | |
| **VOS** | | | | | | 96.77 | | | | | |
| **GAN-Synthesis** | | | | | | N/A | | | | | |
| **OE** | 98.13 | 98.35 | 98.41 | 97.67 | 98.16 | 98.66 | 98.53 | 98.82 | 99.16 | 99.26 | 99.39 |
| **Energy + FT** | 99.76 | 99.81 | 99.88 | 99.90 | 99.91 | 99.95 | 99.97 | 99.98 | 99.99 | 99.99 | 99.99 |
| **WOOD** | 99.19 (<0.01) | 99.39 (<0.01) | 99.55 (<0.01) | 99.62 (<0.01) | 99.82 (<0.01) | 99.78 (<0.01) | 99.80 (<0.01) | 99.85 (<0.01) | 99.89 (<0.01) | 99.91 (<0.01) | 99.92 (<0.01) |
| **SEE-OoD** | **99.86** (<0.01) | **99.97** (<0.01) | **99.97** (<0.01) | **100.0** (0.0) | **100.0** (0.0) | **99.97** (0.0) | **99.97** (0.0) | **99.99** (0.0) | **100.0** (0.0) | **100.0** (0.0) | **100.0** (0.0) |

## E.2 REGIME II EXPERIMENTAL RESULTS

### E.2.1 FASHIONMNIST WITHIN-DATASET EXPERIMENT

Table 15: TPR for Regime II FashionMNIST *Within-Dataset* OoD detection

| TNR | Method | Number of training OoD samples for **SELECTED** OoD class (i.e. class 8) | | | | | | | | | | |
|-----|--------|------|------|------|------|------|------|------|------|------|------|------|
| | | 4 | 8 | 16 | 32 | 64 | 128 | 256 | 512 | 1024 | 2048 | 4096 |
| 95% | **OE** | 37.15 | 39.40 | 41.93 | 42.76 | 46.59 | 49.84 | 51.46 | 51.91 | 53.06 | 53.83 | 53.95 |
| | **Energy + FT** | **69.40** | 70.08 | 71.23 | 70.35 | 70.38 | 69.77 | 70.15 | 72.64 | 76.09 | 72.50 | 70.32 |
| | **WOOD** | 50.07 (3.01) | 52.08 (0.29) | 56.60 (1.43) | 59.92 (1.18) | 65.88 (3.94) | 62.18 (0.74) | 62.98 (1.18) | 62.57 (1.49) | 65.33 (0.81) | 68.00 (0.13) | 66.15 (2.30) |
| | **SEE-OoD** | 48.28 (6.41) | **71.27** (7.74) | **97.98** (1.99) | **96.82** (3.24) | **98.83** (1.52) | **97.25** (2.70) | **97.93** (2.76) | **100.0** (0.0) | **100.0** (0.0) | **99.08** (1.06) | **99.18** (1.09) |
| 99% | **OE** | 12.06 | 11.52 | 13.91 | 16.68 | 20.54 | 22.00 | 23.85 | 25.60 | 27.75 | 29.87 | 34.89 |
| | **Energy + FT** | **49.61** | **50.23** | 51.97 | 53.81 | 56.04 | 55.23 | 55.59 | 57.04 | 58.90 | 56.30 | 55.64 |
| | **WOOD** | 26.23 (3.18) | 38.43 (0.99) | 41.30 (0.80) | 46.05 (0.57) | 49.45 (0.77) | 50.07 (0.39) | 50.72 (0.86) | 51.83 (0.36) | 53.83 (0.40) | 53.30 (0.37) | 53.52 (0.72) |
| | **SEE-OoD** | 15.08 (4.88) | 33.80 (5.97) | **85.97** (8.54) | **86.75** (9.67) | **94.02** (7.64) | **94.50** (4.97) | **95.65** (5.77) | **100.0** (0.0) | **100.0** (0.0) | **97.18** (3.06) | **97.62** (2.51) |

Table 16: AUROC for Regime II FashionMNIST *Within-Dataset* OoD detection

| Method | Number of training OoD samples for **SELECTED** OoD class (i.e. class 8) | | | | | | | | | | |
|--------|------|------|------|------|------|------|------|------|------|------|------|
| | 4 | 8 | 16 | 32 | 64 | 128 | 256 | 512 | 1024 | 2048 | 4096 |
| **OE** | 84.00 | 84.09 | 84.73 | 83.77 | 83.67 | 83.72 | 83.21 | 84.20 | 83.88 | 83.86 | 82.33 |
| **Energy + FT** | 92.61 | 92.56 | 93.08 | 92.94 | 92.97 | 92.99 | 93.12 | 93.78 | 94.85 | 94.39 | 92.78 |
| **WOOD** | 88.07 (1.87) | 87.38 (0.23) | 87.67 (0.95) | 89.24 (1.36) | 90.29 (0.16) | 91.74 (1.22) | 89.33 (1.04) | 89.78 (0.67) | 92.35 (1.65) | 93.73 (0.36) | 89.82 (2.25) |
| **SEE-OoD** | **92.90** (0.49) | **96.13** (0.10) | **99.52** (0.17) | **97.76** (1.93) | **99.77** (<0.01) | **99.60** (0.20) | **99.69** (<0.01) | **100.0** (0.0) | **100.0** (0.0) | **99.84** (<0.01) | **99.85** (<0.01) |

### E.2.2 SVHN WITHIN-DATASET EXPERIMENT

Table 17: TPR for Regime II SVHN *Within-Dataset* OoD detection

| TNR | Method | Number of training OoD samples for ***SELECTED*** OoD class (i.e. class 8) | | | | | | | | | | |
|---|---|---|---|---|---|---|---|---|---|---|---|---|
| | | 4 | 8 | 16 | 32 | 64 | 128 | 256 | 512 | 1024 | 2048 | 4096 |
| 95% | **OE** | 57.79 | 61.69 | 62.27 | 67.62 | 71.49 | 76.93 | 76.84 | 78.86 | 81.38 | 80.61 | 83.41 |
| | **Energy + FT** | **75.79** | **79.82** | 81.57 | 85.65 | 88.14 | 91.55 | 90.91 | 91.83 | 92.41 | 90.63 | 91.24 |
| | **WOOD** | 52.04 (0.90) | 57.09 (1.34) | 64.13 (3.48) | 67.44 (1.87) | 75.77 (0.68) | 78.60 (2.08) | 79.98 (1.08) | 84.28 (1.99) | 85.73 (1.18) | 86.25 (1.02) | 88.13 (0.79) |
| | **SEE-OoD** | 46.35 (1.98) | 42.68 (4.90) | **84.64** (2.91) | **92.25** (3.80) | **98.42** (1.32) | **98.96** (0.92) | **99.67** (0.23) | **100.0** (0.0) | **100.0** (0.0) | **100.0** (0.0) | **100.0** (0.0) |
| 99% | **OE** | 16.01 | 16.90 | 18.65 | 20.40 | 26.61 | 31.00 | 33.49 | 37.63 | 39.54 | 42.49 | 45.87 |
| | **Energy + FT** | **41.97** | **48.73** | 51.00 | 58.31 | 64.49 | 70.38 | 69.43 | 73.58 | 75.33 | 72.69 | 73.70 |
| | **WOOD** | 13.50 (0.26) | 16.82 (0.42) | 23.71 (3.72) | 32.44 (0.76) | 48.04 (2.34) | 54.25 (1.66) | 55.90 (3.17) | 62.43 (3.77) | 66.84 (4.21) | 68.14 (0.86) | 69.25 (0.41) |
| | **SEE-OoD** | 12.46 (0.23) | 11.83 (2.56) | **52.57** (4.61) | **65.48** (9.10) | **92.30** (4.72) | **95.13** (5.37) | **98.17** (1.22) | **100.0** (0.0) | **99.95** (<0.1) | **100.0** (0.0) | **99.96** (<0.1) |

Table 18: AUROC for Regime II SVHN *Within-Dataset* OoD detection

| Method | Number of training OoD samples for ***SELECTED*** OoD class (i.e. class 8) | | | | | | | | | | |
|---|---|---|---|---|---|---|---|---|---|---|---|
| | 4 | 8 | 16 | 32 | 64 | 128 | 256 | 512 | 1024 | 2048 | 4096 |
| **OE** | 91.03 | 92.48 | 92.90 | 93.70 | 94.59 | 95.15 | 95.19 | 94.94 | 94.83 | 94.07 | 94.76 |
| **Energy + FT** | **95.25** | **96.13** | 96.41 | 97.20 | 97.70 | 98.15 | 98.16 | 98.14 | 98.25 | 97.63 | 97.76 |
| **WOOD** | 82.89 (0.83) | 86.80 (1.82) | 88.99 (<0.01) | 90.93 (0.13) | 92.25 (0.46) | 94.45 (0.11) | 95.76 (0.52) | 96.10 (0.52) | 96.33 (<0.01) | 96.23 (<0.01) | 96.78 (0.11) |
| **SEE-OoD** | 89.87 (0.52) | 89.53 (0.29) | **97.36** (0.60) | **98.55** (0.52) | **99.63** (<0.01) | **99.77** (0.12) | **99.91** (<0.01) | **100.0** (0.0) | **100.0** (<0.01) | **100.0** (0.0) | **100.0** (<0.01) |

