# OpenReview forum: "SEE-OoD: Supervised Exploration for Enhanced Out-of-Distribution Detection"
_ICLR.cc/2024/Conference — Submitted to ICLR 2024_

### Official Review · Reviewer_8gYg · 2023-10-27

**Soundness:** 3 good
**Presentation:** 3 good
**Contribution:** 3 good
**Rating:** 8
**Confidence:** 4

**Summary:**

The paper is concerned with the robust out-of-distribution (OOD) detection, while maintaing the same level of performance of in-distribution (IND) data. In order to overcome the case when the number of OOD samples is too smalll (which could lead to overfitting), the authors propose a generative adversarial approach that uses real OOD data for supervised generation of synthetic OOD samples and thus could better represent the OOD space. More concretely, they propose a Wasserstein-score-based generative adversarial training
framework where the generator explores OOD space and synthesis virtual outliers with the feedback provided by the discriminator, while the discriminator exploits the generated outliers to separate IND and OOD data distributions. Extensive experiments demonstrates the superiority of the proposed approach in OOD detection when compared with other state-of-the-art approaches.

**Strengths:**

The paper is well-documented, clearly written and easy to follow. The paper provides a theoretical insight in order to demonstrate the effectiveness of the proposed method. The related work section covers the most relevant papers in the field. Experimental validation is convincing and demonstrates the superiority of the proposed approach.

**Weaknesses:**

The idea is not totally new, i.e. the usage of Wasserstein-based-score for OOD detection has been used before (see the WOOD method).

**Questions:**

- I understand that your approach is suitable for low-data regime (especially when the number of OOD samples is low). It would be interesting to visualize the curves in figures 2 and 3 also as a number of several ratios (IND/OOD). How many number of OOD samples do you need to generate in each case?
- Why do you distinguish between two experimental scenarios (balanced vs imbalanced OOD classes)? It is assumed that OOD data is unlabeled. So, it does not matter how many data is in each class. Please clarify this aspect.
- Another aspect which is not clear to me is how do you define the imbalanced regime for OOD, i.e. you mention 'only a few classes are observed'). I thought imbalance refers to different ratios between the samples of OOD classes or with respect to the samples in the IND classes. What is in each case the ratio between majority classes and minority classes?
- If the generative adversarial approach is unconditional, why the case of imbalanced scenario is relevant? OOD data is unlabeled anyway.
- How do you evaluate the quality of the synthetic OOD samples? Some quantitative and qualitative analysis is indicated.

---

> ### Author Response · Authors · 2023-11-17
> **Author Response to Reviewer 8gYg**
>
> Dear Reviewer 8gYg, we are grateful for your detailed feedback. Please see our following point-to-point responses to address your comments.
>
> ## Presentations of Figure 2. and Figure 3.
>
> Thanks for your valuable suggestions about presenting the results. The reason why we choose to use numbers of OoD samples, instead of InD/OoD ratios, as X-axis is because, in practice, the OoD samples at hand are usually very scarce (ex, unseen manufacturing defects, rarely seen obstacles in autonomous driving), whereas the size of InD training data can grow easily. This makes the ratio metric not as informative as numbers. As for the number of OoD samples that we generate, we generate OoD samples in each iteration of the iterative training process and use those to regularize the model. For detailed information, we kindly refer the reviewer to Appendix D.2.2, where $B_G$ denotes the number of OoD samples generated in each iteration.
>
> ## Differences between two regimes
>
> We are glad that you point out that OoD data are unlabeled anyway. We totally agree with you. Although the OoD labels are not used in the model training, the inherent differences between OoD samples from different classes can affect the model performance. And there might be a distribution shift between observed and test OoD samples. The idea here is to control what has been observed by our method in the training time. Under the balanced sampling strategy in regime I, the testing OoD samples resemble part of training OoD samples. However, in practice, people only possess very limited knowledge about OoD samples so it is reasonable that some types of OoD samples are not observed. Under the imbalanced sampling strategy in regime II, the test OoD samples may be different from the observed OoD samples.
>
> ## Further explanation of imbalanced regime
>
> In the imbalanced regime, we deliberately leave one class (i.e., type) of OoD samples unobserved. For example, in the SVHN Within-Dataset experiment, classes 8 and 9 are considered OoD, but during training, only samples from class 8 are observed. We employ this setting to assess the generalizability of our method. For specific details regarding experimental setups and the number of samples included during training, we kindly refer the reviewer to Table 2 and Appendix E.
>
> ## Evaluation of quality of generated outliers
>
> Thank you for highlighting this crucial aspect. In our paper, Theorem 1 and Theorem 2 enable us to offer theoretical guarantees on the quality of generated outliers. We have demonstrated that, theoretically, the generated outliers will not be part of InD clusters. Regarding the experimental setup, we deliberately refrain from controlling the generation of outliers. This choice aligns with the primary goal and contribution of our method, which is to explore OoD spaces. As long as the generated outliers do not become part of InD, our objective is considered accomplished.

---

> ### Comment · Reviewer_8gYg · 2023-11-22
> **Official Comment by Reviewer 8gYg**
>
> After carefully analyzing authors' responses, I consider that my concerns have been satisfactorily addressed. Therefore, and despite other reviewers' opinions, I have decided to maintain my initial rating. Overall, I consider the paper has a contribution, the proposed approach is supported by a mathematical formalism (including proofs of threorems) and the experimental validation is convincing. I agree that including OOD samples in the training process limits somehow the generalization capability of the method, but it is still considered a valid setting.

---

### Official Review · Reviewer_BP77 · 2023-11-01

**Soundness:** 3 good
**Presentation:** 2 fair
**Contribution:** 2 fair
**Rating:** 3
**Confidence:** 4

**Summary:**

In the presented research, the authors introduce a novel generative adversarial training approach rooted in the Wasserstein-score-based
framework. This method facilitates the generator in traversing Out-of-Distribution (OoD) spaces to produce virtual outliers, guided by feedback from the discriminator. Concurrently, the discriminator harnesses these outliers to distinguish between In-Distribution (InD) and OoD data within the designated Wasserstein score space. The study furnishes theoretical validations confirming the method's robustness, highlighting its capacity to seamlessly segregate InD and OoD data, including the synthesized virtual OoD samples. A unique experimental paradigm is unveiled, termed Within-Dataset OoD detection, which offers a more rigorous test for Deep Neural Networks (DNNs) compared to the conventional Between-Dataset OoD differentiation tasks. The efficacy of the proposed technique is further validated through extensive benchmark tests across varied image datasets.

**Strengths:**

1. The paper introduces a novel generative adversarial training scheme that allows the generator to explore Out-of-Distribution (OoD) spaces and generate virtual outliers. This innovative approach enhances the traditional methods of OoD detection by leveraging the power of generative models.

2. A standout feature of this paper is the provision of several theoretical results that back the proposed method. By demonstrating that the
discriminator can achieve perfect separation between In-Distribution (InD) and OoD samples in the Wasserstein score space, the authors solidify the the credibility of their approach.

**Weaknesses:**

1. Evaluation Metrics: The paper seems to overlook certain prevalent evaluation metrics. For instance, the Area Under the Curve (AUC) is a widely accepted metric in many domains, including OoD detection. It would be beneficial to understand the performance of the proposed method under such widely recognized metrics. Furthermore, the reliance on outlier exposure raises questions. Specifically, OoD detection typically aims to identify data points that deviate from the expected distribution, rather than simply identifying outliers. The distinction between these two is subtle but crucial.

2. Generality of Proposed Setting: The proposition of the Within-Dataset OoD detection is undoubtedly a fresh perspective. However, its universal applicability remains a concern. This setting, which treats different classes within the same dataset as OoD with respect to one another, might not capture the diverse and multifaceted nature of real-world OoD scenarios. To many, this approach might appear as an analysis of variations within a single domain rather than a genuine OoD situation.

3. Scalability Concerns: The scalability of the proposed method, especially when confronted with vast and diverse datasets , remains ambiguous.

**Questions:**

Please refer to the above weakness part.

---

> ### Author Response · Authors · 2023-11-17
> **Author Response to Reviewer BP77**
>
> Dear Reviewer BP77, we are grateful for your detailed feedback. Please see our following point-to-point responses to address your comments.
>
> ## Evaluation Metric
>
> We kindly refer the reviewer to our general response.
>
> ## Generality of Within-Dataset Detection Tasks
>
> We would like to thank Reviewer BP77 for pointing this out. As you alluded, the primary objective of introducing this new setting is to assess OoD detection accuracy when InD and OoD samples closely resemble each other. The resemblance between InD and OoD samples introduces additional challenges compared to the conventional between-dataset detection setting.
>
> We agree with your comment on the universal applicability of Within-Dataset detection tasks.
>
> However, in various real-world situations, InD and OoD samples may resemble each other. For instance, in anomaly detection, anomalies can manifest as small deviations from the InD samples (Sun et al., 2023). This insight motivated the Within-Dataset experiments, where OoD samples originate from classes within the same dataset. The goal was to create scenarios where InD and OoD samples come from the same dataset and emphasize our model's ability to effectively detect such cases.
>
> That said, it is crucial to clarify that the within-dataset detection setting serves as an additional metric when compared to other methods. We have demonstrated that our method significantly outperforms baselines in the traditional Between-Dataset setting, particularly in terms of generalizability to test OoD samples not encountered during the training process. This also indicates that our method effectively leverages existing limited OoD samples and explores potential OoD spaces more efficiently compared to the baselines.
>
> ## Scalability Concerns
>
> We kindly refer the reviewer to our general response. Following the settings in Liu et al. (2020) we added one more Between-Dataset experiment, CIFAR10-Texture with more InD classes, and one more baseline OE (Hendrycks et al., 2018), to illustrate the scalability and effectiveness of our method.
>
> In addition, we would like to communicate the following perspective with the reviewer.
>
> ### Are the difficulties of OoD detection tasks determined by the complexities of datasets?
>
> It is critical to point out that the complexity of the InD and OoD datasets is not the only factor that influences the difficulties of OoD detection tasks. According to Liang et al. (2020), Lee et al. (2018), and Liu et al. (2020), one of the central challenges for OoD detection tasks in classification models arises from the fact that DNN models are usually over-confident towards InD samples.
>
> For example, OoD detection on a simple InD dataset like the FashionMNIST Within-Dataset experiment, where the classification accuracy can easily reach 97% (i.e. high predictive confidence for InD data), is not necessarily easier than CIFAR100-Texture Between-Dataset experiments done in Liu et al., 2020, where the classification accuracy is only about 79% (i.e. low predictive confidence). Indeed, this explains why the baseline methods, including OE (Hendrycks et al., 2018), Energy Finetuning (Liu et al. 2020), and VOS (Du et al., 2022) fail to achieve desirable performance on the simpler datasets but can achieve decent performance on those seemingly complex datasets like CIFAR100-Texture according to Liu et al. 2020.
>
> That said, we have included another set of experiments on CIFAR10-Texture experiments to further highlight the benefits of our approach.
>
> Furthermore, from our experiments, we notice that another critical factor influencing the difficulties of OoD detection tasks is the proximity between InD and OoD samples. This observation motivated the Within-Dataset experiments, where OoD samples come from classes within the same dataset. The idea here was to create situations where InD and OoD samples highly resemble each other and highlight the capability of our model to effectively detect such cases.
>
> While in practice, this situation may not be universally applicable, it does represent various real-world situations, such as anomaly detection, where anomalies can manifest as small deviations from the InD samples or their features (Sun et al. 2023).

---

### Official Review · Reviewer_cBKA · 2023-11-01

**Soundness:** 3 good
**Presentation:** 3 good
**Contribution:** 2 fair
**Rating:** 3
**Confidence:** 5

**Summary:**

This paper presents a generative adversarial training method leveraging a Wasserstein-score to enhance Out-of-Distribution (OoD) detection accuracy. The approach simultaneously undertakes data augmentation and exploration using a limited set of OoD samples. Additionally, the study offers theoretical assurances, confirming that the optimal solutions derived from generative model can be statistically realized through adversarial training in empirical scenarios.

**Strengths:**

The method employs a unique exploration strategy to identify regions where the model lacks confidence. By focusing on uncertain regions, SEE-OOD achieves superior performance in detecting OOD samples compared to existing methods. The paper presents extensive experiments and benchmarks to validate the effectiveness of SEE-OOD against other state-of-the-art techniques.

**Weaknesses:**

1. The paper does not provide visualizations of the generated outliers, which could offer more intuitive insights into the model's behavior and decisions.
2. The evaluation metrics employed in the paper miss out on including the Area Under the Receiver Operating Characteristic (AUROC), which is crucial for understanding model performance in classification tasks, especially in OOD detection.
3. While the paper presents results on certain datasets, it would benefit from testing on larger and more diverse datasets to ensure the method's generalizability and robustness.

**Questions:**

Please address the weaknesses I've highlighted above.

---

> ### Author Response · Authors · 2023-11-17
> **Author Response to Reviewer cBKA**
>
> Dear Reviewer cBKA, we are grateful for your detailed feedback. Please see our following point-to-point responses to address your comments.
>
> ## Visualization of OoD samples
>
> Thank you for your question on the interpretability of the generated outliers.
>
> Given that our approach aims at the exploration of OOD spaces, the generated images need not look like observed OOD images. That said, one can add a distance regularization term within our objective to penalize the distance between OoD samples and the generated outliers. With such penalty, the generated images will resemble observed OoD samples. For your interest, we provided two generated images (for the MNIST experiment, where we treat classes 1 and 7 as OoD) with the distance regularization term. We can see meaningful images are generated in these cases, where interpretability is maintained.
>
> - [Example 1](https://github-production-user-asset-6210df.s3.amazonaws.com/82349855/283628983-99d1d7bc-c707-4f95-87bd-c758fb38f200.png)
> - [Example 2](https://github-production-user-asset-6210df.s3.amazonaws.com/82349855/283629133-beaa0b9f-d336-460c-bcef-10b8095d124f.png)
>
> However, by imposing the penalty term, the exploration of OoD spaces will be constrained. Indeed, we found that such a constraint will decrease generalization power. This observation also agrees with existing literature showing that such a regularization term led to reduced generalizability (Wang et al. 2021, Liu et al. 2020). Therefore, we decided to remove the penalty term in our final formulation.
>
> ## Missing AUROC metrics
>
> We kindly refer the reviewer to our general response.
>
> ## Experiments are limited
>
> We kindly refer the reviewer to our general response. In addition, we would like to communicate the following perspectives with the reviewer.
>
> ### Are the difficulties of OoD detection tasks determined by the complexities of datasets?
>
> It is critical to point out that the complexity of the InD and OoD datasets is not the only factor that influences the difficulties of OoD detection tasks. According to Liang et al. (2020), Lee et al. (2018), and Liu et al. (2020), one of the central challenges for OoD detection tasks in classification models arises from the fact that DNN models are usually over-confident towards InD samples.
>
> For example, OoD detection on a simple InD dataset like the FashionMNIST Within-Dataset experiment, where the classification accuracy can easily reach 97% (i.e. high predictive confidence for InD data), is not necessarily easier than CIFAR100-Texture Between-Dataset experiments done in Liu et al., 2020, where the classification accuracy is only about 79% (i.e. low predictive confidence). Indeed, this explains why the baseline methods, including OE (Hendrycks et al., 2018), Energy Finetuning (Liu et al. 2020), and VOS (Du et al., 2022) fail to achieve desirable performance on the simpler datasets but can achieve decent performance on those seemingly complex datasets like CIFAR100-Texture according to Liu et al. 2020.
>
> That said, we have included another set of experiments on CIFAR10-Texture experiments to further highlight the benefits of our approach.
>
> Furthermore, from our experiments, we notice that another critical factor influencing the difficulties of OoD detection tasks is the proximity between InD and OoD samples. This observation motivated the Within-Dataset experiments, where OoD samples come from classes within the same dataset. The idea here was to create situations where InD and OoD samples highly resemble each other and highlight the capability of our model to effectively detect such cases. While in practice, this situation may not be universally applicable, it does represent various real-world situations, such as anomaly detection, where anomalies can manifest as small deviations from the InD samples or their features (Sun et al. 2023).

---

> > ### Comment · Reviewer_cBKA · 2023-11-21
> >
> > After a comprehensive examination of the authors' responses to the raised questions and the feedback provided by other reviewers, I have made the decision to revise my initial score to "reject." As I highlighted in my initial review, I continue to believe that the paper could significantly improve by incorporating more intuitive insights into the model's behavior and decision-making process, as well as by conducting testing on larger and more diverse datasets. At the moment, it is still uncertain whether the model demonstrates satisfactory performance.

---

### Official Review · Reviewer_Qajt · 2023-11-02

**Soundness:** 2 fair
**Presentation:** 3 good
**Contribution:** 2 fair
**Rating:** 3
**Confidence:** 5

**Summary:**

This paper studies OOD detection relying on real OOD samples in training. It proposes a method to generate more OOD samples in training based on Wasserstein-score-based generative adversarial learning. Experiments show that the proposed method can achieve good OOD detection performance, given more OOD samples in training.

**Strengths:**

- The method can achieve good performance on the experimented settings and datasets, which matches the assumptions that seeing real OOD samples can help OOD detection in testing.
- The experiments studied the two settings with balanced or imbalanced OOD samples in training.
- The paper is written clearly.

**Weaknesses:**

- Some arguments and claims are the paper are impervious or unfaithful.
    - The real OOD samples are not used in many methods because the OOD samples are unknown/unpredictable in training. Handling OOD detection without OOD samples in training is a more general and real setting, instead of a drawback. It is reasonable to use some OOD samples to perform “outlier exposure”. However, the related works are not discussed in the paper.

- The experiments are limited.
    - The experiments only cover the simple datasets and settings as shown in Table 2. The “within-dateset” setting and the used datasets are simple. And the dataset used in “between-dataset” setting are also not complex enough the validate the methods. That’s also why the proposed method can easily achieve very high performance after seeing real OOD samples. More “between-dataset” settings should be considered as more recent OOD detection papers, such as (Liu et al., 2020).
    - The compared methods are limited and unfair. Many OOD detection methods using OOD samples in training are not discussed and compared, such as the “outlier exposure” based methods (“Deep Anomaly Detection with Outlier Exposure”).
    - Some recent strong OOD detection methods are not discussed or compared, such as
Non-Parametric Outlier Synthesis, ICLR 2023.
How to Exploit Hyperspherical Embeddings for Out-of-Distribution Detection?, ICLR 2023.
    - Many strong OOD detection do not use OOD samples in training (in the more general and real setting) but have strong modeling. But it is straightforward to introduce the known OOD samples in the training process easily. The authors should consider this case and conduct comparisons, especially considering the experimented settings and datasets are very simple.

**Questions:**

Please address the questions mentioned in the weakness, especially those about experiments.

---

> ### Author Response · Authors · 2023-11-17
> **Author Response to Reviewer Qajt**
>
> Dear Reviewer Qajt, we are grateful for your detailed feedback. Please see our following point-to-point responses to your comments.
>
> # Some arguments are impervious or unfaithful
>
> Based on your comments, we added more explanations to clarify our paper's goal and to justify the rationale and advantages of our method. We kindly refer the reviewer to our "general response".
>
> # Experiments are limited
>
> ## 1. Datasets are simple
>
> We address this question in our "general response". Specifically, we followed your suggestion and included one more Between-Dataset experiment on the CIFAR10 and Texture dataset introduced in Liu et al. 2020. In addition, we would like to communicate the following perspective with the reviewer.
>
> ### Are the difficulties of OoD detection tasks determined by the complexities of datasets?
>
> It is critical to point out that the complexity of the InD and OoD datasets is not the only factor that influences the difficulties of OoD detection tasks. According to Liang et al. (2020), Lee et al. (2018), and Liu et al. (2020), one of the central challenges for OoD detection tasks in classification models arises from the fact that DNN models are usually over-confident towards InD samples.
>
> For example, OoD detection on a simple InD dataset like the FashionMNIST Within-Dataset experiment, where the classification accuracy can easily reach 97% (i.e. high predictive confidence for InD data), is not necessarily easier than CIFAR100-Texture Between-Dataset experiments done in Liu et al., 2020, where the classification accuracy is only about 79% (i.e. low predictive confidence). Indeed, this explains why the baseline methods, including OE (Hendrycks et al., 2018), Energy Finetuning (Liu et al. 2020), and VOS (Du et al., 2022) fail to achieve desirable performance on the simpler datasets but can achieve decent performance on those seemingly complex datasets like CIFAR100-Texture according to Liu et al. 2020.
>
> That said, we have included another set of experiments on CIFAR10-Texture experiments to further highlight the benefits of our approach.
>
> Furthermore, from our experiments, we notice that another critical factor influencing the difficulties of OoD detection tasks is the proximity between InD and OoD samples. This observation motivated the Within-Dataset experiments, where OoD samples come from classes within the same dataset. The idea here was to create situations where InD and OoD samples highly resemble each other and highlight the capability of our model to effectively detect such cases.
>
> ## 2. Baselines are missing
>
> We kindly refer the reviewer to our "general response".
>
> ## 3. Recent strong OoD detection methods are not compared
>
> We want to thank the reviewer for pointing out these two excellent papers which came out in May 2023. However, we have to respectfully mention that it is a rare request to conduct a comparison study with the methods that are officially published four months prior to the submission deadline.
>
> ## 4. It is straightforward to introduce OoD samples in the training process
>
> Please refer to our responses in the "general response", where we reiterate the main contributions of this paper. Secondly, we respectfully disagree with Reviewer Qajt's comment suggesting that incorporating OoD samples into the training process is straightforward. It is, instead a very challenging task that needs to be done carefully. There are three primary reasons for this:
>
> - Methods that solely rely on existing OoD samples without exploring OoD spaces, such as Energy Finetuning (Liu et al., 2020), WOOD (Wang et al., 2021), and OE (Hendrycks et al., 2018), fail to detect OoD samples not encountered in the training process, leading to poor generalizability. The main pitch of our method is to effectively use observed OoD samples to perform iterative exploration of OoD spaces. Our experimental results in the imbalanced regime substantiate this claim.
> - Existing methods that exploit OoD samples are susceptible to overfitting when OoD samples are scarce in the training dataset, as demonstrated in our experiments. In such cases, augmenting the limited OoD samples is desirable, particularly in the original image space rather than the feature space. This is because feature spaces may exhibit high overlap, even for images that appear vastly different.
> - We agree with Reviewer Qajt that many methods that do not use OoD samples have strong modeling. However, their modeling approach is mainly based on operations (including exploration) within InD spaces (Du et al. 2022; Ming et al. 2022; Lee et al. 2017). In these cases, there is no straightforward method to incorporate OoD samples during the training stage. Introducing OoD samples naively using score regularization is likely to give rise to the aforementioned two problems.

---

### Author Response · Authors · 2023-11-17
**General Response**

We would like to first thank all the reviewers for carefully reading our paper and providing detailed feedback. There are three common comments that reviewers have raised: (1) Validity of utilizing real OoD samples (2) Limitations of experiments, and (3) Missing AUROC metrics. So, we would like to address those questions first as below.

## Validity of introducing real OoD samples

We concur with Reviewer Qajt and Reviewer BP77 that OoD detection in the absence of actual OoD samples is a commonplace scenario in practical applications and has been investigated more in the literature. However, in many applications, domain experts possess valuable, albeit limited, empirical knowledge about what OoD samples may look like. For example:

- in automatic defect classification tasks for manufacturing quality inspection, domain experts often possess engineering knowledge of critical defect types that may have not yet been observed within the training data, such as for products using new materials or tools (see Sun et al., 2023, for example)
- in facial recognition tasks, researchers have shown that it is reasonable to treat similar but non-human faces (e.g. Chimpanzees) or low-quality faces as OoD samples (Yu et al., 2020)
- in OoD detection for autonomous driving perception (Nitsch et al., 2021), rarely-seen objects and obstacles should be used as existing OoD samples to enhance system safety.

In such settings, developing methodologies that aim to optimally leverage these limited OOD samples can bring critical advantages. Indeed, this is the central motivation and contribution of our paper: **providing a systemic framework for incorporating limited OoD samples for improved OoD detection accompanied by strong (and first-of-a-kind) generalization guarantees for unseen OOD samples.**

## Limitations of experiments

To address comments from Reviewer Qajt, Reviewer cBKA, and Reviewer BP77 about our experiments, we added one more Between-Dataset experiment following the suggestion from Reviewer Qajt. Specifically, we experimented with our method and baselines on the widely-used CIFAR10-Texture InD/OoD pair that was used in the suggested paper by Liu et al., 2020. In addition, we added one more suggested baseline, Outlier Exposure (Hendrycks et al., 2018), to further highlight the advantageous properties of our approach. We report the TPR under 95% TNR and the AUROC in the following table (the formatting is: TPR/AUROC).

| Number of OoD Samples | 128               | 256               | 512               | 1024              | 2048              |
| --------------------- | ----------------- | ----------------- | ----------------- | ----------------- | ----------------- |
| OE                    | 61.03 / 93.07     | 64.50 / 93.89     | 69.32 / 95.01     | 80.60 / 96.87     | 94.05 / 98.70     |
| Energy + FT           | 84.73 / 96.65     | 88.86 / 97.72     | 92.91 / 98.66     | 97.69 / 99.48     | 99.17 / 99.87     |
| WOOD                  | 94.53 / 98.70     | 95.80 / 99.06     | 98.19 / 99.65     | 99.41 / 99.77     | 99.89 / 99.93     |
| SEE-OoD               | **99.80 / 99.98** | **100.0 / 100.0** | **100.0 / 100.0** | **100.0 / 100.0** | **100.0 / 100.0** |

It can be seen that our method outperforms all benchmarks. This superiority is specifically significant with limited OoD samples. This result again confirms the benefits set forth by our proposed approach by exploring OoD spaces.

## Missing AUROC metrics

Based on the comments from Reviewer cBKA and Reviewer BP77 about the missing metric AUROC, we added AUROC comparisons between our model and the baselines. The results also consistently show the significant advantages of our method over all baselines.

The tables of AUROC values for all experiments can be found in Appendix E of our revised paper. Also, one example is provided in the Table above.

## Reference

1. Liu et al. 2020. Energy-based out-of-distribution detection.
2. Nitsch et al. 2021. Out-of-distribution detection for automotive perception
3. Wang et al. 2021. WOOD: Wasserstein-based out-of-distribution detection.
4. Yu et al. 2020. Out-of-distribution detection for reliable face recognition
5. Sun et al. 2023. A continual learning framework for
   adaptive defect classification and inspection
6. Hendrycks et al. 2018. Deep anomaly detection with outlier exposure
7. Li et al. 2022. Vos: Learning what you don’t know by virtual outlier synthesis.
8. Ming et al. 2023. How to exploit hyperspherical embeddings
   for out-of-distribution detection?
9. Lee et al. 2018. A simple unified framework for detecting
   out-of-distribution samples and adversarial attacks
10. Lee et a. 2018. raining confidence-calibrated classifiers
    for detecting out-of-distribution samples
11. Liang et al. 2020. Enhancing the reliability of out-of-distribution image
    detection in neural networks.

---

### Meta-Review · Area_Chair_ZVFP · 2023-11-30

**Metareview:**

This paper targets out-of-distributions detection and assumes an out-of-distribution sample is available during training in addition to in-distribution data. They then employ a generative model to approximate the out-distribution sample and claim that doing so enables better exploration and exposes their model to a larger set of possible test out-distributions.

Points of improvement highlighted by reviewers include the fact that the proposal is tailored to the out-of-distribution sample observed during training and variations around it, which doesn't ensure a broad coverage of all the possible out-distributions that might occur. The evaluation is also limited since it doesn't cover other approaches that access out-of-distribution data during training, and some recent work with competing approaches is ignored. The considered datasets are also somewhat small scale, and they focus on out-distributions given by new classes unobserved during training or completely different datasets, which is arguably not so challenging as more subtle variations such as quality perturbations and synthetic or adversarially generated data. I would recommend expanding the evaluation prior to publication.

**Justification For Why Not Higher Score:**

The evaluation requires improvement to provide clear evidence of the authors's claims.

**Justification For Why Not Lower Score:**

N/A

---

### Decision · Program_Chairs · 2024-01-16

Reject